# Dopamine and opioid systems interact within the nucleus accumbens to maintain monogamous pair bonds

**Shanna L Resendez[1,2]\*, Piper C Keyes[3], Jeremy J Day[4], Caely Hambro[3], Curtis J Austin[3], Francis K Maina[5], Lori N Eidson[6], Kirsten A Porter-Stransky[3,7], Natalie Nevárez[3], J William McLean[4], Morgan A Kuhnmuench[3], Anne Z Murphy[6], Tiffany A Mathews[5], Brandon J Aragona[1,3]\***

[1]Neuroscience Graduate Program, University of Michigan, Ann Arbor, United States; [2]University of North Carolina, Chapel Hill, United States; [3]Department of Psychology, University of Michigan-Ann Arbor, Ann Arbor, United States; [4]Department of Neurobiology, University of Alabama at Birmingham, Birmangham, United States; [5]Department of Chemistry, Wayne State University, Detroit, United States; [6]Neuroscience Institute, Georgia State University, Atlanta, United States; [7]Department of Human Genetics, Emory University, Atlanta, United States

**\*For correspondence:**
shanna_resendez@med.unc.edu
(SLR); aragona@umich.edu (BJA)

**Competing interests:** The authors declare that no competing interests exist.

**Abstract** Prairie vole breeder pairs form monogamous pair bonds, which are maintained through the expression of selective aggression toward novel conspecifics. Here, we utilize behavioral and anatomical techniques to extend the current understanding of neural mechanisms that mediate pair bond maintenance. For both sexes, we show that pair bonding up-regulates mRNA expression for genes encoding D1-like dopamine (DA) receptors and dynorphin as well as enhances stimulated DA release within the nucleus accumbens (NAc). We next show that D1-like receptor regulation of selective aggression is mediated through downstream activation of kappa-opioid receptors (KORs) and that activation of these receptors mediates social avoidance. Finally, we also identified sex-specific alterations in KOR binding density within the NAc shell of paired males and demonstrate that this alteration contributes to the neuroprotective effect of pair bonding against drug reward. Together, these findings suggest motivational and valence processing systems interact to mediate the maintenance of social bonds.

## Introduction

The ability to maintain meaningful social bonds is a critical component of human health and mental well being, yet the neural capacity to maintain such relationships is not well understood. The socially monogamous prairie vole (*Michrotus ochrogaster*) presents an ideal animal model to study the neural correlates of social bond maintenance because, unlike most mammals (*Kleiman, 1977*), prairie voles form selective and enduring attachments to their mating partner (*Aragona et al., 2009*). In both the field and laboratory, the maintenance of these bonds is associated with the expression of selective aggression towards novel conspecifics as well as selective affiliation with the mating partner (i.e., mate guarding) (*Carter and Getz, 1993*). Importantly, the expression of selective aggression provides a robust and reliable assay that can be utilized in a laboratory setting to deconstruct neural signaling pathways involved in the regulation of social bond maintenance.

To date, laboratory studies have identified that the expression of selective aggression, and therefore pair bond maintenance, requires the activation of both D1-like dopamine (DA) and kappa-

**eLife digest** The bond between parents is one of the most important social relationships that humans have. Prairie voles are one of the few other mammals whose individuals also form long-term social bonds after having offspring together, so they have frequently been used to study the brain mechanisms that underlie such bonding. However, most previous studies have focused only on how the bond between a pair of mating partners is formed: little is known about how this bond is then maintained over months and years.

When a prairie vole forms a bond with a mate, it will then aggressively reject other prairie voles. This "selective aggression" only happens once a social bond between two mating prairie voles is formed, so this behavior can be used as a proxy to confirm that the social bond exists.

In order to study how prairie voles maintain bonds with a mate, Resendez et al. tracked what happens in the brain of a prairie vole during selective aggression. The experiments showed that this aggressive behaviour coincides with changes in gene expression and brain chemistry that make it unpleasant for a prairie vole to be exposed to voles that are not its partner. For male prairie voles – but not females – these changes only happened if the female mating partner became pregnant during the cohabitation period.

The changes that occur in the brain as a result of bonding with a partner also mean that drugs that are normally addictive are no longer pleasant and rewarding to the prairie vole. Indeed, forming a social bond between mating animals alters the brain in similar ways to the effects produced by addictive drugs. Thus, in a sense, each member of the mating pair becomes 'addicted' to their partner.

The results presented by Resendez et al. also have implications for humans. They suggest that having a strong social support network is a powerful way of preventing casual drug use from developing into compulsive drug addiction. This may also mean that positive social relationships could help to treat people with drug addiction problems.

opioid receptors (KORs) within the nucleus (NAc) shell as blockade of either one of the receptors attenuates aggressive rejection of novel conspecifics (*Aragona et al., 2006*; *Resendez et al., 2012*). Thus, regulation of pair bond maintenance requires neural systems that code evaluation of salient environmental stimuli as well as those that are important for the generation of motivational states (*Resendez and Aragona, 2013*). Interestingly, in other animal models, these receptor systems have been shown to directly interact at the molecular level (*Gerfen et al., 1990*; *Carlezon et al., 1998*) as well as in the transition between motivational states (*Chartoff et al., 2016*). However, it is unknown if similar interactions occur in the regulation of pair bond maintenance. This study therefore endeavored to examine pair bond induced neural plasticity within the DA and dynorphin/KOR systems as well as how these systems interact to mediate the expression of selective aggression, a well established indicator of a fully established pair bond.

Given that activation of KORs is associated with aversive states (*Mucha and Herz, 1985*; *Pfeiffer et al., 1986*; *Shippenberg and Herz, 1986*; *Bals-Kubik et al., 1989*), we first determined if activation of NAc KORs prior to pairing with a novel social stimulus is sufficient to tag a recently encountered social stimulus as aversive. Next, to assess how the establishment of a pair bond alters both motivational (DA) and aversive (dynorphin/KOR) processing systems, we conducted extensive anatomical, neurochemical, and functional comparisons within the striatum of male and female prairie voles. In total, we conducted mRNA expression analysis (RT-qPCR), protein binding measurements (receptor autoradiography), and measures of DA concentration (fast-scan cyclic-voltammetry) to identify sex-specific alterations within the DA and dynorphin/KOR systems of pair bonded voles. We next utilized site-specific behavioral pharmacology to examine interactions between NAc shell D1-like and KORs in the expression of selective aggression. Finally, in male prairie voles, we show that pair bonding, but not other social manipulations, decreases the rewarding properties of the psychostimulant amphetamine and that this attenuation requires the activation of NAc shell KORs. In total, the present study demonstrates that the development of a pair bond is underpinned by sex-specific modifications in motivational (DA/D1) and valence (dynorphin/KOR) processing systems, that

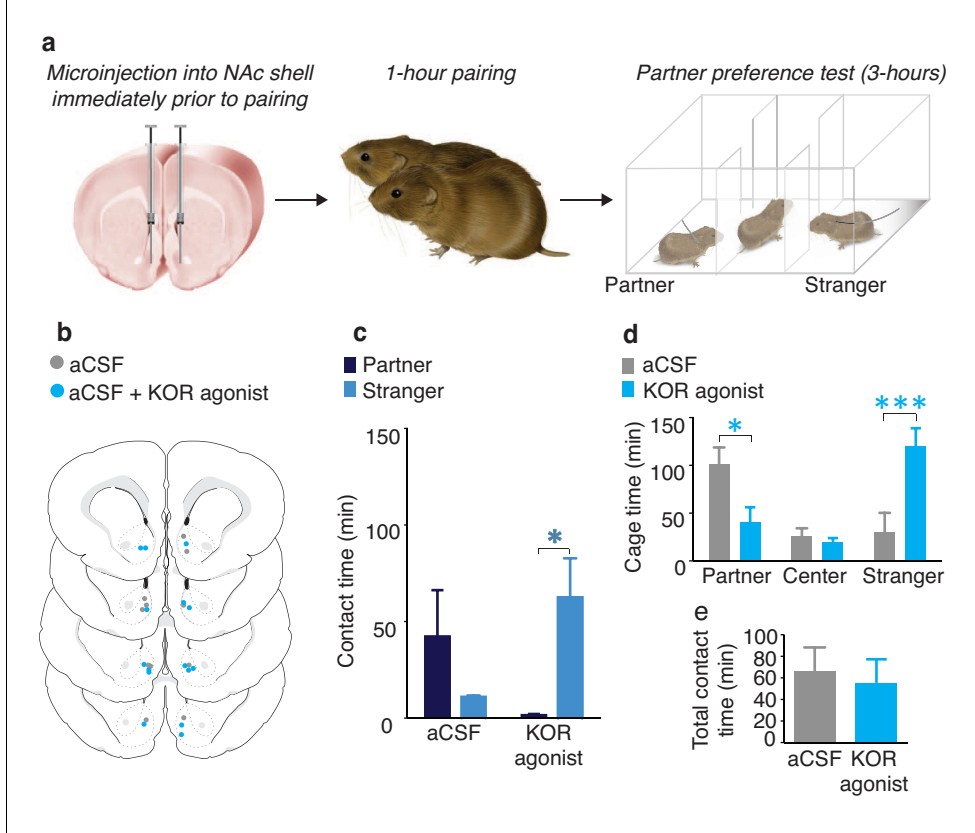

**Figure 1.** NAc shell KORs encode social aversion. (a) Experimental design. (b) Histological verification of injection sites. (c) Control males (aCSF) paired with a female partner for 1 hr showed no social preference or aversion (n = 6). In contrast, activation of NAc shell KORs via site-specific administration of a KOR agonist induced a partner aversion (n = 7). (d) Males that received site-specific injections of the KOR agonist also spent significantly less time in the partner's cage as well as more time in the chamber containing the stranger. (e) There was no difference in total contact time between the two groups. Summary data are presented as mean ± SEM. *p<0.05, **p<0.005.

The following figure supplement is available for figure 1:

**Figure supplement 1.** Social and grooming behavior during 1-hr cohabitation.

these systems interact to mediate selective aggression in both sexes, and that male specific alterations in the dynorphin/KOR system buffers against the rewarding properties of amphetamine.

## Results

### KORs within the NAc shell encode social aversion

Activation of NAc shell KORs is required for the expression of selective aggression by pair bonded voles (*Resendez et al., 2012*); however, the psychological processes that underlie the expression of this behavior are not well understood. In other rodent species, activation of these receptors has been shown to induce aversion as well as mediate avoidance behaviors (*Land et al., 2008*; *Al-Hasani et al., 2015*). For example, pairing of a previously neutral stimulus with either an aversive experience that results in KOR activation, such as stress, or with direct pharmacological activation of these receptors results in the avoidance of that stimulus during subsequent encounters (*Land et al., 2008*). Given the known relationship between aversive processing of environmental stimuli, avoidance behaviors (*Boren et al., 1959*; *D'Amato et al., 1967*), and KOR activation, we hypothesized that one mechanism in which NAc shell KORs mediate social avoidance behaviors is through the encoding of novel social stimuli as aversive.

To determine if activation of NAc shell KORs during a social encounter results in social avoidance behaviors, we utilized a modified version of the partner preference paradigm (*Figure 1a*). Specifically, we employed a social pairing condition (1 hr cohabitation with an opposite sex conspecific) that is insufficient to produce a preference for the familiar partner over an unfamiliar conspecific (the stranger). A lack of a preference for either social stimulus is indicated by equivalent amounts of time spent with the partner and stranger during the social choice test, suggesting that both social stimuli are of equal valences. As expected, a Wilcoxon signed rank sum test for non-parametric data demonstrated that control males treated with aCSF did not show a preference for either individual ($W_{(5)}$ = 33, z = −0.97, p=0.33) (*Figure 1b,c*). Conversely, male subjects that were administered a KOR agonist (1 µg U50,488) into the NAc shell immediately prior to pairing with the female partner avoided the female that had been paired with KOR activation and therefore displayed a robust preference for contact with the novel female that had not been previously paired with NAc shell KOR (Wilcoxon signed rank sum test, $W_{(6)}$ = 32.5, z = -2.56, p=0.01) (*Figure 1b,c*). In addition to differences in direct contact time, activation of NAc shell KORs prior to pairing with a social stimulus also resulted in differences in the duration of time spent in each stimulus chamber (two-way ANOVA, ($F_{(2,36)}$ = 7.07, p=0.003). Specifically, males that received administration of a KOR agonist avoided the chamber containing the partner (Bonferroni's post hoc test, p=0.02) and spent more time in the chamber occupied by the stranger (p=0.0006) (*Figure 1d*). Finally, control males and males receiving site-specific administration of the KOR agonistdid not differ in total contact time (time spent with partner + time spent with stranger) (t-test, $t_{(11)}$ = 0.35, p=0.73), indicating that reduced contact with the partner did not result from a general decrease in motivation for social contact (*Figure 1e*). Both groups of male subjects also did not differ in affiliative social behavior or grooming behavior during the 1 hr cohabitation (*Figure 1—figure supplement 1*).

Together, these data suggest that activation of KORs within the NAc shell induces social avoidance behaviors, potentially through the assignment of negative valence onto a previously neutral social stimulus. Given that activation of NAc shell KORs is required for the expression of selective aggression in pair bonded voles, it is possible that KOR activation within the NAc shell mediates pair bond maintenance by assigning social stimuli other than the mating partner with a negative valence signal. We therefore conducted our next series of experiments to determine how pair bonding alters neural systems involved in the regulation of selective aggression to promote pair bond maintenance.

## Pair bond induced alteration in mRNA expression within the ventral striatum

Sexually naïve prairie voles find social novelty rewarding and will readily approach and interact with novel conspecifics. In stark contrast, a pair bonded vole will avoid and aggressively reject this same social stimulus, suggesting that they find social stimuli—other than their mating partner or offspring—to be aversive (*Resendez and Aragona, 2013*). We therefore hypothesized that this behavioral transformation is mediated by an up-regulation of neural systems that regulate the expression of selective aggression, such as both the D1-like receptor and dynorphin/KOR systems within the ventral region of the striatum. Thus, to determine if non-pair bonded (sibling housed) and pair bonded (2 weeks cohabitation with an opposite-sex conspecific) voles differ in the expression level of mRNA for genes that encode proteins involved in the regulation of selective aggression, we utilized RT-qPCR to compare the level of mRNAs related to the DA and dynorphin/KOR systems. For all groups, comparisons were made within the dorsal and ventral striatum (i.e., NAc).

Extensive cohabitation with a mating partner predominately altered the expression of mRNA for genes that code for proteins associated with pair bond maintenance. Specifically, within the ventral striatum, t-test results show that pair bonded males and females showed higher levels of mRNA for the gene encoding dynorphin (*Pdyn*) (Male: $t_{(24)}$ = 2.26, p=0.03; $t_{(26)}$ = 3.05, p=0.005), the endogenous ligand for KORs. They also showed elevated levels of mRNA expression for the gene encoding D1-like receptors (*Drd1*) (t-test; Male: $t_{(24)}$ = 2.86, p=0.009; Female: $t_{(28)}$ = 3.42, p=0.002) (*Figure 2a and c*). For paired males, similar elevations in mRNA for the gene that encodes KOR (*Oprk1*) also occurred; however, due to high levels of variability in the expression of this gene, elevations in Oprk1 mRNA did not significantly differ from non-paired males (*Table 1*). Only moderate elevations in *Oprk1* mRNA levels occurred in females and this elevation failed to reach significance (*Table 1*). Finally, differences in the expression of mRNA for *Drd1* and *Pdyn* were not identified within the dorsal striatum indicating that these changes are specific to the ventral region of the striatum (*Table 2*).

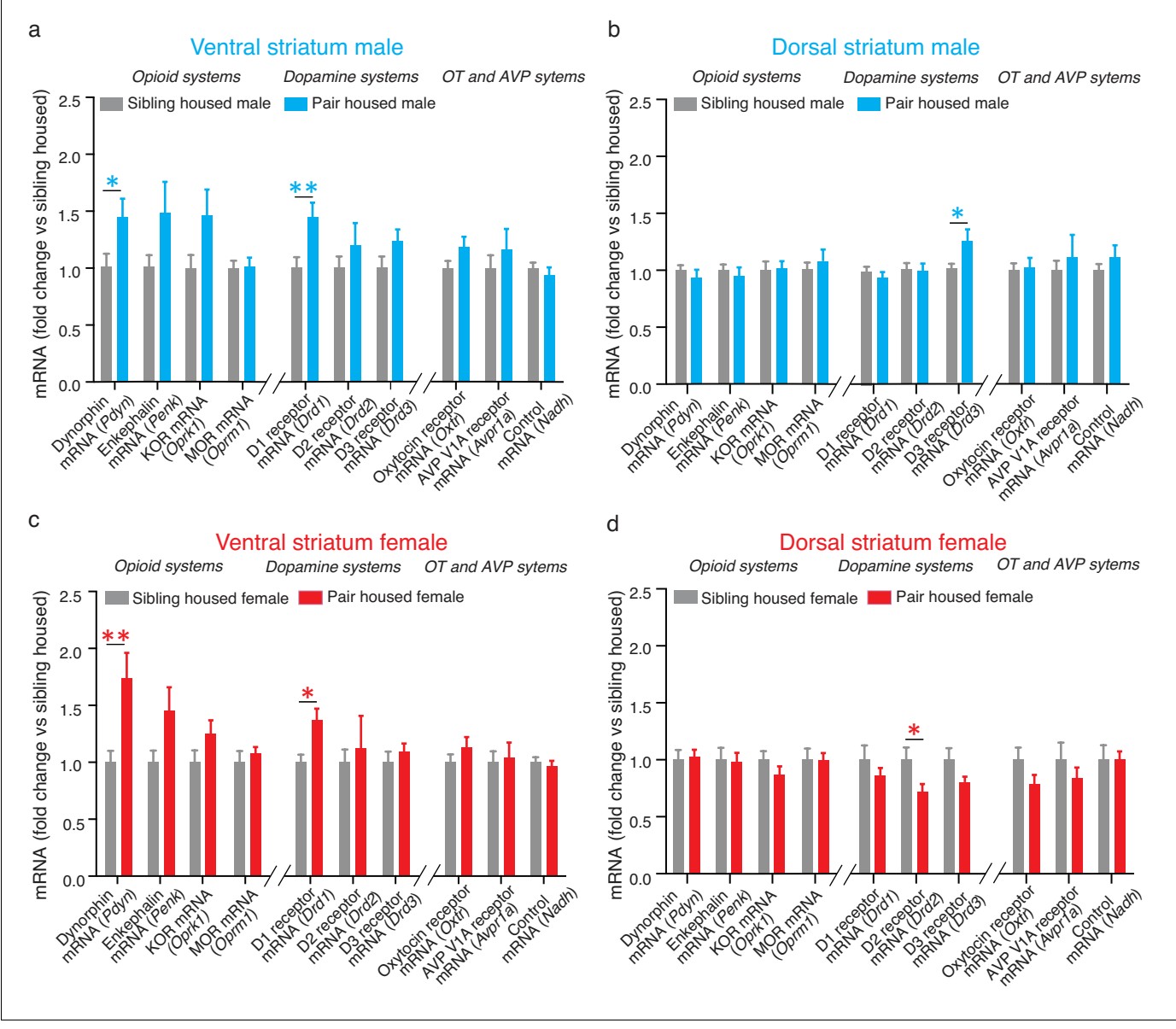

**Figure 2.** Pair bonding alters mRNA expression within the NAc. (a) Pair bonding increased the expression of *Drd1* and *Pdyn* mRNA within the VS of males (n = 15/group). (b) Pair bonding decreased *Drd3* mRNA expression within the DS of males (n = 15–16/group). (c) Similar to males, pair bonding increased the expression of *Drd1* and P*dyn* within the VS of females (n = 6–23/group). (d) Pair bonding significantly decreased *Drd2* mRNA within the DS of paired females (n = 16/group). *p<0.05, **p<0.005.

To next determine if differences in expression differences following 2 weeks of male-female cohabitation were specific to genes that encode proteins involved in the regulation of pair bond maintenance, we also examined the expression of genes that encode proteins that regulate social behaviors associated with pair bond formation. Within the NAc, these proteins include D2-like DA receptors, mu-opioid receptors (MORs), and the oxytocin receptor. Following 2 weeks of male-female cohabitation, differences in the expression of genes related to pair bond formation (*Drd2, Penk/Oprm1, Oxtr*) were not found within the ventral striatum of pair bonded voles indicating that the differences identified above are specific to neural systems that regulate pair bond maintenance (*Table 1*). Also in contrast to the above findings, sex-specific alterations in the expression of genes related to pair bond formation were identified within the dorsal striatum. Specifically, compared to non-paired subjects, pair bonded males had higher levels of *Drd3* mRNA (t-test; $t_{(26)} = 2.34$, p=0.03)

**Table 1.** Non-significant statistics for mRNA comparisons in the ventral striatum.

| | Sex | |
| --- | --- | --- |
| Gene | Male | Female |
| Pdyn | NA | NA |
| Penk | $t_{(24)}$ = 1.80, p = 0.09 | $t_{(26)}$ = 1.92, p = 0.07 |
| Oprk1 | $t_{(24)}$ = 1.99, p = 0.06 | $t_{(11)}$ = 0.36, p = 0.72 |
| Oprm1 | $t_{(24)}$ = 0.13, p = 0.90 | $t_{(26)}$ = 0.70, p = 0.49 |
| Drd1 | NA | NA |
| Drd2 | $t_{(24)}$ = 0.10, p = 0.33 | $t_{(37)}$ = 1.57, p = 0.13 |
| Drd3 | $t_{(24)}$ = 1.58, p = 0.13 | $t_{(26)}$ = 0.75, p = 0.46 |
| Oxtr | $t_{(24)}$ = 1.72, p = 0.10 | $t_{(37)}$ = 1.12, p = 0.27 |
| Avpr1a | $t_{(24)}$ = 0.82, p = 0.43 | $t_{(37)}$ = 0.25, p = 0.81 |
| Nadh | $t_{(23)}$ = 1.23, p = 0.23 | $t_{(28)}$ = 0.79, p = 0.44 |

while pair bonded females had higher levels of *Drd2* mRNA (t-test; $t_{(26)}$ = 2.12, p=0.04) (*Figure 2b, d*). No other differences were identified within the dorsal striatum.

Overall, the above findings are consistent with the proposed mechanism that the establishment of a pair bond is associated with region specific alterations in neural systems that regulate selective aggression. However, an up-regulation in the expression of mRNA is not always indicative of an increase in protein levels. We therefore utilized receptor autoradiography to examine pair bond induced differences in KOR binding density within the striatum. We focused on KORs in the present study because it has previously been shown that pair bonding increases the expression of D1-like receptors specifically within the ventral striatum (*Aragona et al., 2006*).

## Sex specific alterations in KOR binding

To determine whether pair bonding alters striatal KOR density, KOR binding densities were compared between non-paired (i.e., same-sex sibling housed) and pair bonded prairie voles (i.e., 2 weeks male-female cohabitation) of both sexes. Compared to non-paired (sibling housed) males, a two-way ANOVA indicated that pair bonded males had lower levels of striatal KOR binding density ($F_{(1,120)}$ = 17.51, p=0.0001; *Figure 3a,b*). Further examination of pair bond induced alterations in KOR binding density within the striatum of males revealed that the decrease in KOR binding was specific to the ventral region of the NAc shell (Bonferroni's post hoc test, p=0.01; *Figure 3b*;

**Table 2.** Non-significant statistics for mRNA comparisons in the dorsal striatum.

| | Sex | |
| --- | --- | --- |
| Gene | Male | Female |
| Pdyn | $t_{(26)}$ = 0.80, p = 0.43 | $t_{(26)}$ = 0.21, p = 0.83 |
| Penk | $t_{(26)}$ = 0.56, p = 0.58 | $t_{(26)}$ = 0.13, p = 0.90 |
| Oprk1 | $t_{(26)}$ = 0.17, p = 0.86 | $t_{(26)}$ = 1.19, p = 0.24 |
| Oprm1 | $t_{(26)}$ = 0.63, p = 0.53 | $t_{(26)}$ = 0.05, p = 0.96 |
| Drd1 | $t_{(26)}$ = 1.15, p = 0.26 | $t_{(26)}$ = 0.88, p = 0.39 |
| Drd2 | $t_{(26)}$ = 0.18, p = 0.86 | NA |
| Drd3 | NA | $t_{(26)}$ = 1.69, p = 0.10 |
| Oxtr | $t_{(26)}$ = 0.20, p = 0.84 | $t_{(26)}$ = 1.50, p = 0.15 |
| Avpr1a | $t_{(26)}$ = 0.56, p = 0.58 | $t_{(26)}$ = 0.85, p = 0.40 |
| Nadh | $t_{(26)}$ = 0.97, p = 0.34 | $t_{(26)}$ = 0.005, p = 0.10 |

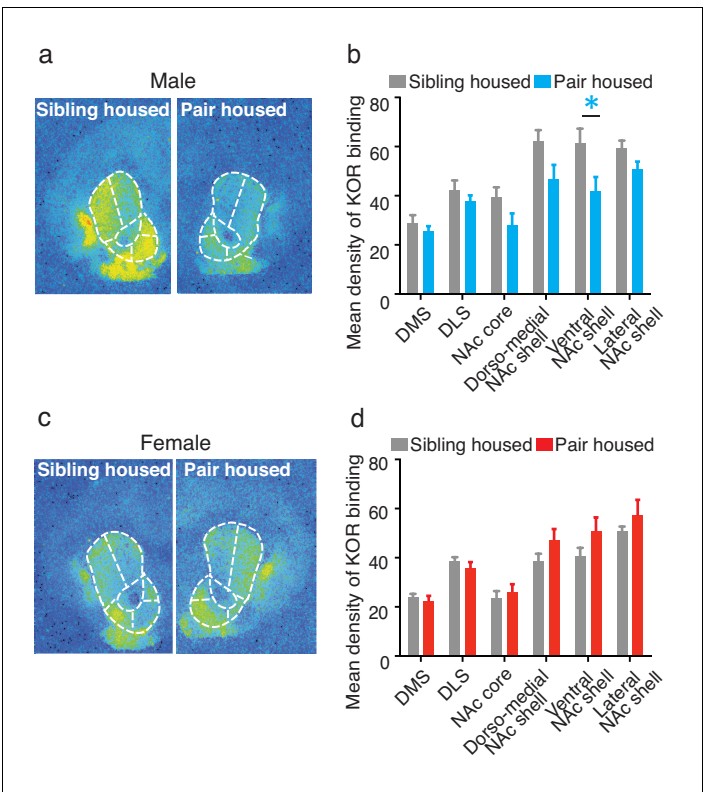

**Figure 3.** Pair bonding alters KOR binding within the striatum of males. (a,b) Pair bonding decreased KOR binding in the dorso-medial and ventral NAc shell of males (n = 11/group). (c,d) There was no significant effect on KOR binding density in females (n = 10/group). Summary data are presented as mean ± SEM. *p<0.05.
The following figure supplement is available for figure 3:

**Figure supplement 1.** Sex differences in KOR binding density before and after pair bonding.

_Table 3_), the region of the striatum where KORs act to regulate selective aggression (_Resendez et al., 2012_) and mediate aversion (_Al-Hasani et al., 2015_).

In contrast to paired males, significant alterations in KOR binding density following the establishment of a pair bond were not identified in females ($F_{(1,108)}$ = 3.50, p>0.06; _Figure 3c,d_) suggesting that pair bonding induces sex-specific alterations in KOR binding density. We therefore next compared KOR binding density between males and females before and after the establishment of a pair bond. Prior to the establishment of a pair bond, a two-way ANOVA indicated that non-paired (sibling housed) males have significantly higher levels of KOR binding density within the striatum compared to non-paired females (two-way ANOVA, $F_{(1,114)}$ = 38.14, p=0.0001). Specifically, non-paired males had significantly higher levels throughout the NAc, including the NAc core (Bonferroni's post

**Table 3.** Non-significant statistics for comparisons of KOR binding density in paired versus unpaired males.

| Striatal sub-region | Bonferonni's _post hoc_ test |
| --- | --- |
| Dorso-medial striatum | p>0.99 |
| Dorso-lateral striatum | p>0.99 |
| NAc core | p = 0.41 |
| Nac dorso-medial shell | p = 0.07 |
| NAc lateral shell | p = 0.99 |

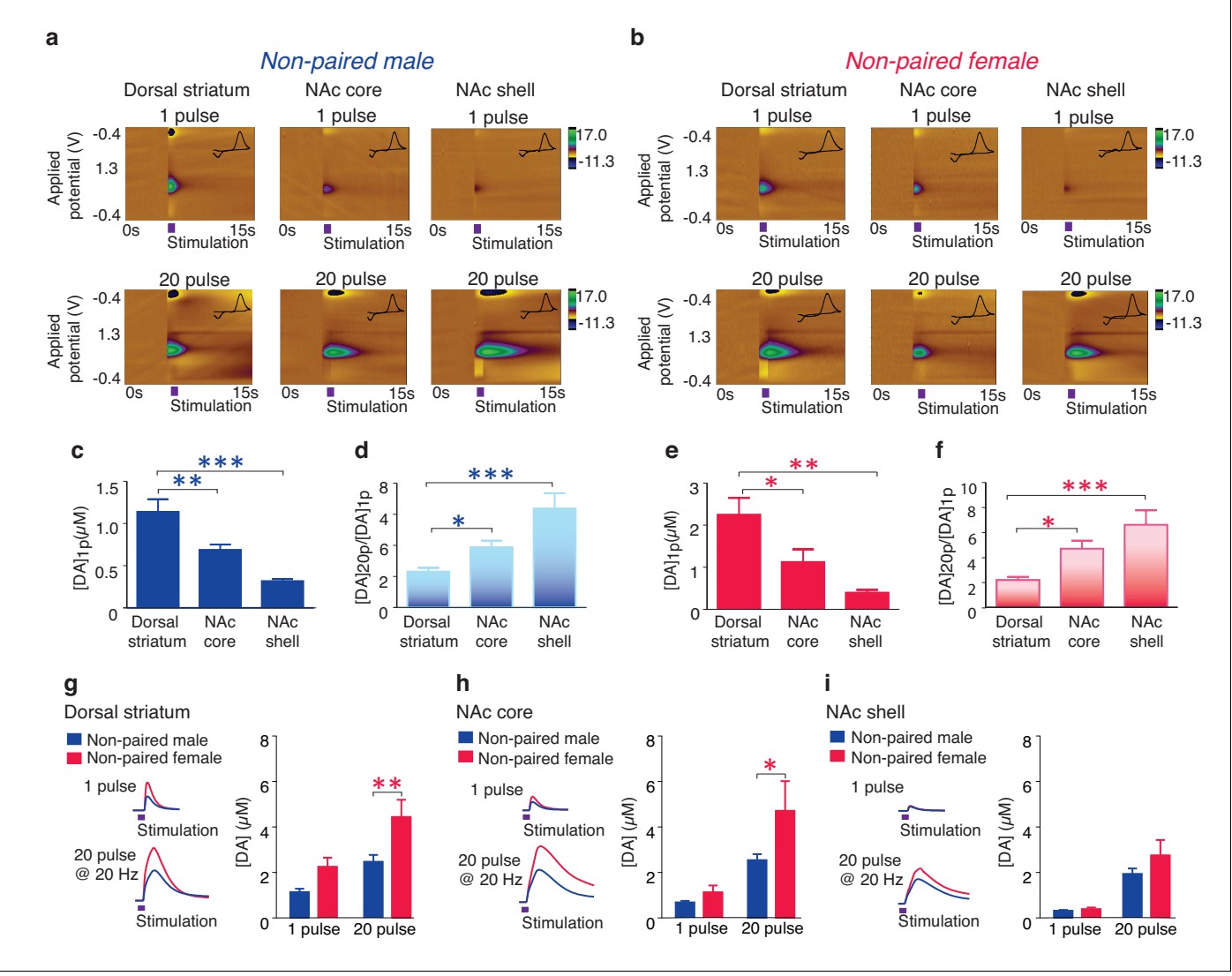

**Figure 4.** Striatal DA transmission in non-pair bonded prairie voles. (a,b) Representative color plots of DA transmission throughout the striatum of (a) male and (b) female prairie voles. (c,e) A 1-pulse depolarizing stimulation evokes the greatest magnitude of DA release within the dorsal striatum and the magnitude of this release decreases along a dorsal to ventral gradient within the striatum of (c) males and (e) females. (d,f) An inverse relationship is seen with burst facilitation as the greatest ratio of DA release occurs within the NAc shell, an intermediate ratio occurs within the NAc core, and the lowest ratio occurs within the dorsal striatum of (d) males and (f) females. (g–i) Compared to male prairie voles, a 20-pulse stimulation evokes a greater magnitude of DA release within the (g) dorsal striatum and the (h) NAc core of females. (i) No sex difference in DA transmission occurred within the NAc shell. Summary data are presented as mean ± SEM. *p<0.05, **p<0.005, ***p<0.0005.

hoc test; p=0.02), the dorso-medial region of the NAc shell (p=0.0001), and the ventral region of the NAc shell (p=0.005). Interestingly, these sex differences in KOR binding density were not identified in pair bonded males and females as males no longer showed higher levels in KOR binding density (two-way ANOVA, $F_{(1,114)}$ = 0.36, p=0.55) (*Figure 3—figure supplement 1*). Together, these data suggest that pair bonding results in a reduction in KOR binding density within the NAc of male, but not female prairie voles.

## Prairie vole DA release dynamics

Previous studies have established an essential role for the activation of NAc shell D1-like receptors in the expression of social behaviors important for pair bond maintenance (*Aragona et al., 2009*). These receptors are primarily of the low-affinity sub-type (*Richfield et al., 1989*) and their activation

requires high concentrations of DA to be released, such as that which occurs during burst firing of DA neurons (*Gonon, 1997*; *Cheer et al., 2007*). Given that activation of D1-like receptors require high concentrations of DA release and that selective aggression is only expressed in the pair bonded state, we predicted that pair bonded voles would have greater concentrations in DA release specifically within the NAc shell. To compare DA release dynamics between non-bonded and pair bonded voles, we utilized fast-scan cyclic-voltammetry (FSCV) to measure real-time DA release across the striatum. However, given that striatal DA release properties are unknown in this species, we first conducted a detailed characterization of DA release dynamics within the prairie vole striatum (*Figure 4a,b*).

Consistent with other mammals (*Jones et al., 1995*; *Calipari et al., 2012*), the concentration of striatal DA release evoked by a single pulse stimulation ($[DA]_{1p}$) significantly decreased along a dorsal to ventral gradient (one-way ANOVA; Male: $F_{(2,30)} = 17.28$, p<0.000; Female: $F_{(2,27)} = 8,57$, p=0.001). Post hoc Tukey comparisons revealed that both the NAc core (Male: p=0.009; Female: p=0.04) and the NAc shell (Male: p=0.000; Female: p=0.001) had significantly lower levels of

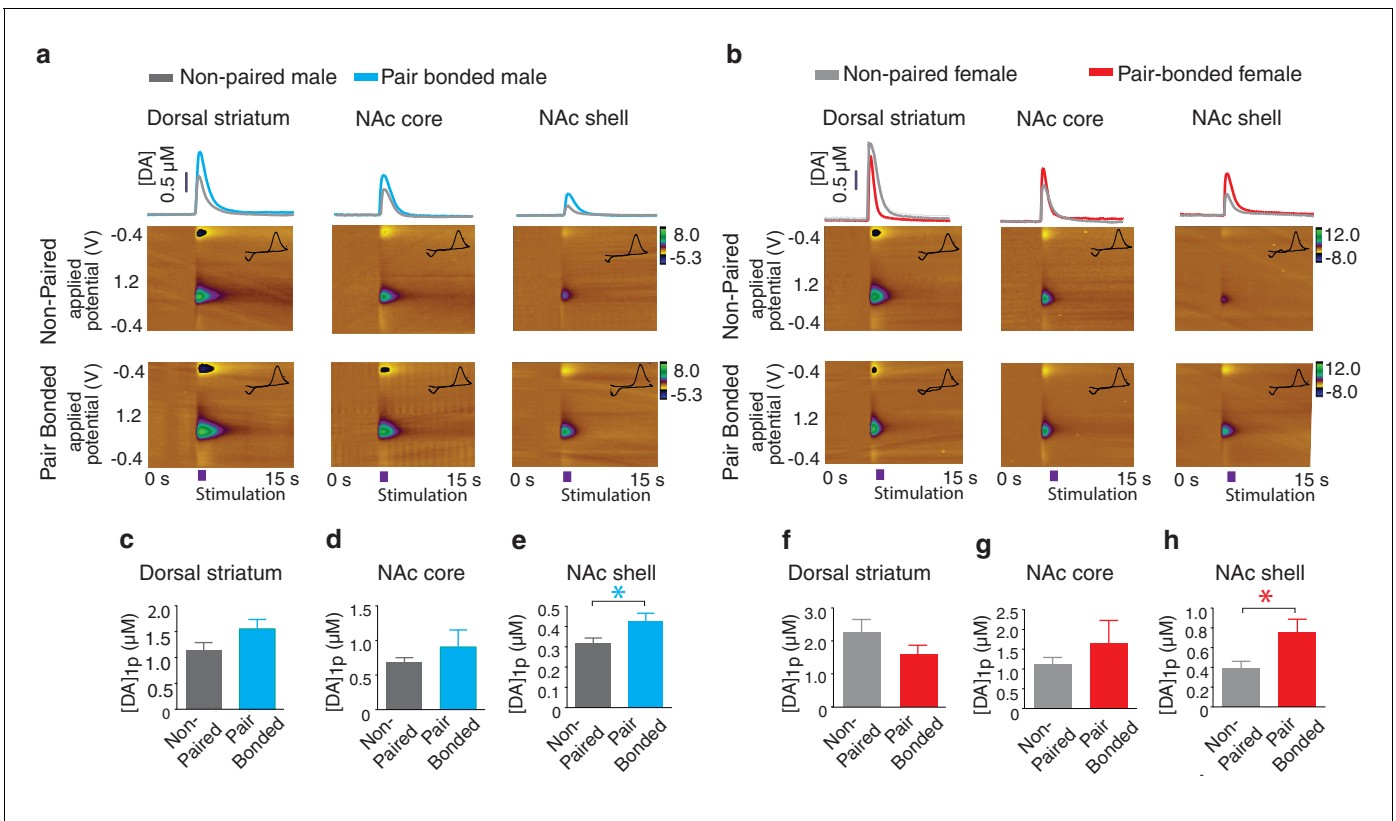

**Figure 5.** Pair bonding enhances NAc shell DA release. (**a,b**) Representative color plots of stimulated DA release following a 1-pulse depolarizing stimulation in (**a**) male and (**b**) female subjects. For both sexes, top row shows representative color plots for non-paired subjects and bottom row shows representative color plots for pair bonded subjects. (**c,d**) Pair bonding had no effect on DA transmission within the (**c**) dorsal striatum (n = 11–12/group) or (**d**) the NAc core of males (n = 10/group). (**e**) Within the NAc shell, a 1-pulse stimulation resulted in significantly greater DA release within the NAc shell of paired males compared to non-paired male controls (n = 9–10/group). (**f,g**) There was no difference in peak DA release between non-paired and pair bonded females following a 1-pulse stimulation within the (**f**) dorsal striatum (n = 8–11/group) or (**g**) the NAc core (n = 8–9/group). (**h**) Similar to males, a 1-pulse depolarizing stimulation resulted in a greater level of DA release within of the NAc shell of paired females compared to non-paired females (n = 7–8/group). Summary data are presented as mean ± SEM. *p<0.05.

The following figure supplements are available for figure 5:

**Figure supplement 1.** Sex differences in striatal dopamine release following the establishment of a pair bond.

**Figure supplement 2.** Pair bonding increases selective aggression in both male and female prairie voles.

stimulated DA release compared to the dorsal striatum (*Figure 4c,e*). Also, consistent with other species (*Zhang et al., 2009*), the magnitude of DA release following stimulation parameters that evoke burst-like firing of DA neurons, such as an extra-physiological 20-pulse stimulation ([DA]$_{20p}$), differed across striatal sub-regions, with the most robust impact occurring within the NAc shell, a moderate impact within the NAc core, and a minimal effect within the dorsal striatum (one-way ANOVA, Male: $F_{(2,28)}$ = 11.22, p=0.0003; Female: $F_{(2,25)}$ = 10.60, p<0.000). This effect is represented by the greatest ratio of evoked DA release ([DA]$_{20p}$/[DA]$_{1p}$) within the NAc shell (Tukey post hoc test; Male: p=0.0002; Female: p=0.0004), an intermediate ratio within the NAc core (Tukey post hoc test; Male: p=0.02; Female: p=0.03), and the lowest ratio within the dorsal striatum (*Figure 4d,f*). In addition, a two-way ANOVA followed by Bonferroni's *post hoc* tests identified significant sex-differences in striatal DA release following a 20-pulse stimulation within the dorsal striatum ($F_{(1,10)}$ = 5.25, p=0.002; p=0.03, *Figure 4g*) as well as the NAc core ($F_{(1,34)}$ = 4.05, p=0.05; p=0.05, *Figure 4h*), but not the NAc shell ($F_{(1,32)}$ = 1.77, p=0.1, *Figure 4i*). Similar sex differences have previously been reported in other species (*Walker et al., 2000*). Overall these results suggest that general striatal DA release patterns appear to be conserved among rodents.

## Pair bond induced enhancement of DA release

Next, to test the hypothesis that pair bonded voles have elevated DA release specifically within the NAc shell of the striatum, electrically evoked DA release was compared across striatal sub-regions of pair bonded and non-pair bonded voles (*Figure 5a,b*). As predicted, t-test comparisons indicated that pair bonding significantly increased peak DA release within the NAc shell of pair bonded voles (Male: t$_{(17)}$ = 2.44, p=0.03; Female: t$_{(13)}$ = 2.48, p=0.03), but not other regions of the striatum (Dorsal striatum male: t$_{(21)}$ = 0.09, p=1.75; Dorsal striatum female: t$_{(17)}$ = 1.26, p=0.22; NAc core male: t$_{(18)}$ = 0.87, p=0.40; NAc core female: t$_{(15)}$ = 0.73, p=0.48) (*Figure 5c–h*). Additionally, although pair bonding significantly elevated NAc shell DA release in both sexes, the average percent increase was lower in males (34%) compared to females (99%) (*Figure 5e,h*). Direct comparisons of peak DA release between pair bonded males and females indicated that pair bonded females had significantly higher levels of DA release within the NAc shell compared to that of pair bonded males (*Figure 5—figure supplement 1*). This sex difference in pair bond induced changes in DA transmission is unlikely due to initial sex differences in NAc shell DA release as differences in DA release within the NAc shell were not identified between non-paired male and females (*Figure 4*). Moreover, given that the release of DA is required for the activation of D1-like receptors that mediate selective aggression and that displays of selective aggression are qualitatively larger in pair bonded males than females (*Figure 5—figure supplement 2*), we initially expected increases in DA transmission to be greater in males.

One possible explanation underlying sex differences in pair bond induced alterations in DA transmission is that the influence of fecundity on pair bond strength differs between males and females (*Resendez et al., 2012*; *McCracken et al., 2015*). More specifically, for pair bonded males, but not females, the strength of the pair bond, as indicated by the magnitude of selective aggression displayed toward intruders, is dependent on pair fecundity. We therefore tested the hypothesis that variations in pair bond induced increases in DA release between males and females were associated with reproductive success.

## Pair fecundity influences DA transmission dynamics in a sexually dimorphic manner

Prior to FSCV recordings of stimulated DA release in striatal slices, fecundity of the pair was assessed by determining the stage of pregnancy following 2 weeks of male-female cohabitation. Briefly, the stage of pregnancy was determined as previously described by measuring the average neonatal weight of the offspring, with larger neonatal weights indicating shorter delays in the onset of pregnancy (*Curtis, 2010*; *Resendez et al., 2012*). Measures of neonatal weight were then used to classify the pairs as either optimally (mating and fertilization occurring within 48–72 hr of pairing) or sub-optimally (delay in establishment of pregnancy) pregnant (*Resendez et al., 2012*).

Following 2 weeks of cohabitation with an opposite sex partner, males from optimally pregnant pairs showed significantly higher levels of aggression than males from sub-optimally pregnant pairs (t-test, t$_{(9)}$ = 2.54, p=0.03) (*Figure 6a*). In contrast, reproductive status had no impact on pair bond

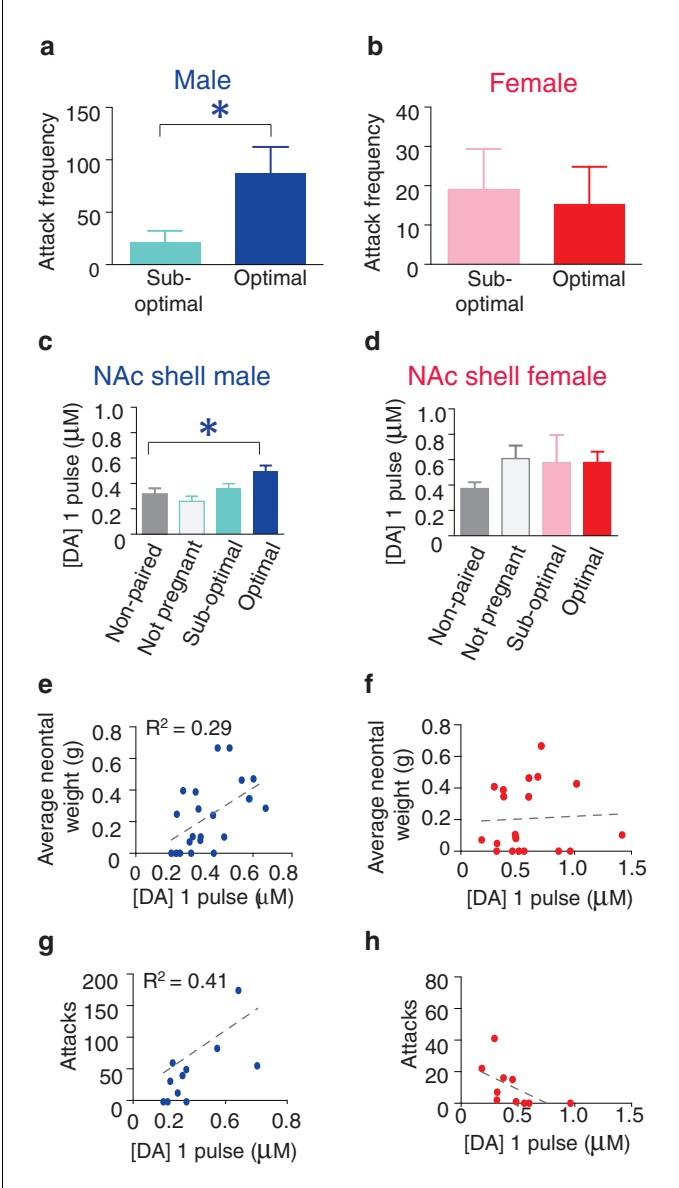

**Figure 6.** Relationship between striatal DA release and characteristics of pair bonding. (**a**) Pair bond induced increases in selective aggression was dependent on fecundity in males as males from optimally pregnant pairs were more aggressive than males from sub-optimally pregnant pairs (n = 5–6/group). (**b**) Conversely, pregnancy optimality had no effect on attack frequency in females (n = 4–7/group). (**c**) Within the NAc shell, males whose females were optimally pregnant showed significantly greater levels of DA release (n = 4–18/group). (**d**) In contrast, for females, there was no difference in peak DA release within the NAc shell between non-paired females and paired females categorized by their reproductive status (n = 5–13/group). (**e**) Among pair bonded males, neonatal weight (an established indicator of gestational stage) was positively correlated with peak DA release within the NAc shell (n = 23). (**f**) However, there was no relationship between peak DA release and reproductive status in paired females. (**g,h**) Finally, in relation to attack frequency, there was a positive correlation between peak DA release and attack frequency within the within the NAc shell of (**g**) paired males (n = 8), but no such relationship was identified among paired females (n = 10). Summary data are presented as mean ± SEM. *$p<0.05$.

The following figure supplements are available for figure 6:

**Figure supplement 1.** Sex differences in selective aggression by fecundity.

*Figure 6 continued on next page*

*Figure 6 continued*

**Figure supplement 2.** Impact of fecundity on dopamine transmission within the dorsal striatum and NAc core of paired male and female prairie voles.

strength in paired females as females from optimally and sub-optimally pregnant pairs did not differ in levels of selective aggression (t-test, $t_{(9)}$ = 0.24, p=0.82) (*Figure 6b*). Moreover, direct comparisons of aggression levels among males and females from sub-optimally and optimally pregnant pairs indicates that sex differences in the magnitude of selective aggression that is displayed toward an intruder depends on pair fecundity (two way ANOVA, $F_{(1,37)}$ = 8.32, p=0.007). Specifically, although paired males are generally more aggressive than paired females, when aggression levels were further compared by fecundity classification, only males from optimally pregnant pairs showed significantly higher levels of selective aggression than females (Bonferroni's post hoc test, optimally pregnant: p=0.01, sub-optimally pregnant: p>0.99) (*Figure 6—figure supplement 1*). Thus, pair fecundity strongly influences pair bond strength in male, but not female prairie voles and only males from optimally pregnant pairs shower higher levels of aggression than paired females. We next determined if fecundity also resulted in sex specific alterations in DA transmission.

Similar to measures of selective aggression, fecundity influenced DA transmission within the NAc shell in a sex-specific manner. Specifically, examination of DA release properties in relation to the pairs reproductive status revealed that only males from optimally pregnant pairs showed significantly greater elevations in NAc shell DA release compared to non-paired males (one-way ANOVA, $F_{(3,39)}$ = 0.29, p=0.05; Dunnett's post hoc test, p=0.04) (*Figure 6c*). In contrast to paired males, reproductive status did not influence NAc shell DA transmission dynamics of paired females (one-way ANOVA, $F_{(3,31)}$ = 1.67, p=0.20). Rather, females of all reproductive categories (not pregnant, sub-optimally pregnant, or optimally pregnant) showed modest elevations in stimulated DA release compared to non-paired females (*Figure 6d*). Thus, it is possible that sex-differences in the magnitude of change in NAc shell DA transmission dynamics results from paired females showing elevations in stimulated DA release regardless of reproductive status, whereas only males from optimally pregnant pairs (7 out of 22 total pairs) had enhanced DA transmission within the NAc shell.

To further explore the relationship between pair fecundity and NAc shell DA transmission dynamics, we next examined the relationship between neonatal weight and DA release. For paired males, the magnitude of stimulated DA release was positively correlated with neonatal weight, with fecundity accounting for nearly 30% of the variation (linear regression, $R^2$ = 0.29, $F_{(1,20)}$ = 8.16, p=0.01) (*Figure 6e*). Conversely, there was no relationship between fecundity and the magnitude of stimulated DA release within the NAc shell DA of paired females (linear regression, $R^2$ = 0.32, $F_{(1,8)}$ = 3.77, p=0.09 (*Figure 6f*). Together, these data suggest that reproductive status alters DA transmission within the NAc shell of paired voles in a sex-specific manner. Moreover, these effects are primarily localized to the NAc shell as paired voles categorized by their reproductive status did not differ in DA transmission dynamics within the NAc core or dorsal striatum (*Figure 6—figure supplement 2*). However, it should be noted that despite a lack of overall differences in DA transmission within the dorsal striatum based on the categorization of pairs by pregnancy, a comparatively modest correlation between DA transmission and pregnancy was found within the dorsal striatum of males (*Figure 6—figure supplement 2*). Nonetheless, these data suggest that fecundity exerts sex-specific effects on DA transmission dynamics in pair bonded prairie voles. Given the identified relationship between fecundity and selective aggression and fecundity and DA transmission, we next examined if variations in DA transmission within the NAc shell contribute to variable levels of selective aggression in paired males.

Prior to measures of stimulated DA release, resident intruder test were administered to male and female subjects by placing a same-sex intruder into the test subjects home cage. Following the completion of behavioral testing, stimulated DA release was measured within the striatum and the frequency of attack behavior was quantified by an experimentally blind observer. These measures were subsequently utilized to assess the relationship between the magnitude of stimulated DA release and the degree of selective aggression displayed toward a resident intruder.

For pair bonded males, the intensity of aggression directed toward a resident intruder was positively correlated with NAc shell DA release, accounting for over 40% of the variation (linear

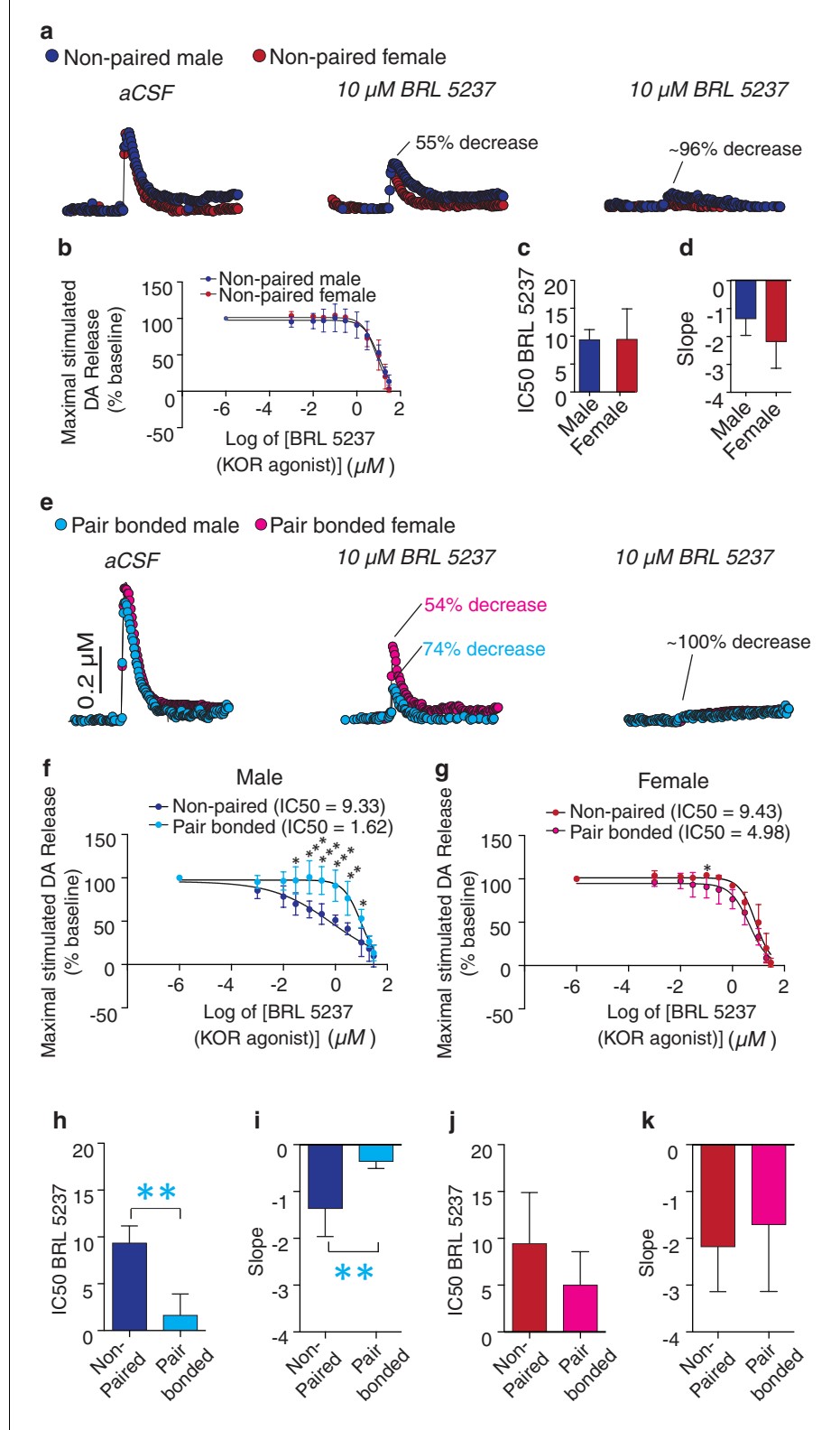

**Figure 7.** Pair bonding increases KOR modulation of NAc shell DA release in male prairie voles. (a,b) Similar to other species, bath application of a KOR agonist decreases DA release in the NAc shell of male and female prairie voles. (c,d) Non-paired males and females did not differ in (c) the IC50 of BRL 5237 (a KOR agonist) or (d) in the slope of the dose response curve. (e) Pair bonding induced sex-specific alterations in KOR modulation of DA

*Figure 7 continued on next page*

*Figure 7 continued*

transmission within the NAc shell (f,g) Following the establishment of a pair bond, KOR mediated decrease of stimulated DA release was enhanced within the NAc shell of (f) males (n = 3–5/group), but not (g) females. (h,i) Compared to non-paired males, pair bonding significantly decreased (h) the IC50 of BRL 5237 as well as (i) the slope of the dose response curve in paired males. (j,k) Pair bonding did not alter KOR mediated DA transmission in females (n = 3–4/group). Summary data are presented as mean ± SEM. **p<0.005.

The following figure supplements are available for figure 7:

**Figure supplement 1.** Comparison of the prairie vole KOR protein sequence to other rodent species and humans.

**Figure supplement 2.** Sex differences KOR modulation of NAc shell DA release following the establishment of a pair bond.

regression, $R^2$ = 0.41, $F_{(1,9)}$ = 6.32, p=0.03) (*Figure 6g*). In contrast, no relationship between NAc shell DA release and selective aggression was identified in paired females (linear regression, $R^2$ = 0.32, $F_{(1,8)}$ = 3.77, p=0.09) (*Figure 6h*). Additionally, no relationship between stimulated DA release and attack frequency was found within other regions of the striatum for either sex (*Figure 6—figure supplement 2*). Thus, the relationship between stimulated DA release and attack behavior in pair bonded prairie behaviors occurs in a sex and region specific manner. When these data are considered in combination with site-specific pharmacology data demonstrating that activation of D1-like DA receptors specifically within the NAc shell is required for the expression of selective aggression (*Aragona et al., 2006*), they suggest that the degree to which DA transmission dynamics are altered within the NAc shell of paired males may underlie fecundity induced modulation of pair bond strength. In other words, enhancement of DA release would facilitate the activation of low-affinity D1-like receptors, possibly leading to the display of higher levels of aggression by males from optimally pregnant pairs. Moreover, stimulation of D1-like receptors results in the production of dynorphin (*Engber et al., 1992*), the endogenous ligand for KORs (*Chavkin et al., 1982*) and activation of these receptors is also required for the expression of selective aggression (*Resendez et al., 2012*). Therefore, we next examined the possibility of pair bond induced alterations in interactions between these systems.

## Pair bonding alters KOR regulation of DA transmission in a sex-specific manner

Activation of KORs within the NAc reduces DA release within this region (*Britt and McGehee, 2008*). Given that pair bonding altered NAc KOR expression pattern, we next compared KOR modulation of DA transmission within the NAc shell of non-paired and paired voles. Similar to other rodent species (*Britt and McGehee, 2008*), bath application of a KOR agonist (**BRL 5237**) onto striatal slices of non-pair bonded voles reduced stimulated DA release within the NAc shell (*Figure 7a, b*). Similar effects on DA transmission were observed in both non-paired males and females as the concentration response curves did not significantly differ between the sexes (two-way ANOVA, $F_{(1, 5)}$ = 1.59, p=0.26). Moreover, a t-test did not identify significant sex differences in the dose required to achieve a 50% reduction in DA release (IC50: $t_{(5)}$ = 0.03, p=0.98; *Figure 7C*) or in the slope of the concentration response curve ($t_{(5)}$ = 1.28, p=0.26, *Figure 7D*). However, in contrast to other species, a much higher dose of the KOR agonist was needed to achieve a 50% decrease in DA release (IC50 ~ten fold greater compared to other rodent species; [*Britt et al., 2012*]).

The necessity to use higher doses of a KOR agonist in the present study is consistent with our previous findings showing that, compared to other rodents, prairie voles also require about a 10X higher dose of a peripherally administered KOR agonist to achieve significant alterations in KOR-mediated analgesia as well as locomotor activity (*Resendez et al., 2012*). The consistent requirement for higher doses of a KOR agonist to observe either a behavioral or physiological impact in prairie voles suggests potential value in comparing the genetic sequence of the prairie vole KOR to other species that have been used to study KOR pharmacology (e.g., rats, mice, guinea pigs, and humans). Indeed, the prairie vole KOR is distinct from the above-mentioned species as its genetic sequence diverges from that of rats and mice (whose KOR structure is quite homologous) as well as

humans and guinea pigs (whose KORs also share substantial homologies) (*Figure 7—figure supplement 1*). It is also notable to mention that the prairie vole KOR is more similar to that of humans and guinea pigs than that of rats and mice in which most pharmacological studies have been conducted. In total, there are four amino acids that are unique to prairie voles, humans, and guinea pigs (Alanine 28, Serine 186, Aspartic acid 218, Aspartic acid 374), one amino acid that is unique to humans and prairie voles (Isoleucine 232), and fifteen amino acids that are unique to prairie voles, including one residue that is located in the dynorphin binding site (*Rasakham and Liu-Chen, 2011*; *Wu et al., 2012*). It is possible that these genetic differences may partially account for species differences in KOR pharmacology and determining how ligands interact with the prairie vole KOR will be an important future area of study. Nevertheless, the above data demonstrate that activation of KORs within the prairie vole striatum produces the expected decreases in DA transmission.

We next compared KOR modulation of DA transmission within the NAc shell of non-bonded (sibling housed) and pair bonded (2 weeks cohabitation with a mating partner) prairie voles to determine if pair bond induced alterations in KOR protein binding within the NAc shell impact KOR modulation of DA transmission. Similar to anatomical changes, pair bonding robustly altered KOR modulation of DA release within the NAc shell of pair bonded males, while only producing very modest alterations in females (*Figure 7e*). More specifically, in male prairie voles, pair bonding resulted in a leftward shift in the concentration response curve (two-way ANOVA, $F_{(1, 6)} = 15.67$, p=0.008) and significantly larger reductions in stimulated DA release at multiple concentrations of the KOR agonist (Bonferroni's post hoc test; 0.1 μM, p=0.03;. 3 μM, p=0.0007; 1 μM, p=0.0004; 3 μM, p=0.0003; 10 μM, p=0.002; 20 μM, p=0.02) (*Figure 7f*). In contrast, the concentration response curve only slightly differed between paired and non-paired females (two-way ANOVA, $F_{(1, 6)} = 9.21$, p=0.03) with only one resulting in greater reduction in stimulated DA release (Bonferroni's post hoc test; 0.3 μM, p=0.02) (*Figure 7g*, *Table 4*). Overall, these data suggest that alterations in paired males were more dramatic than those that occurred in paired females.

Comparison of the IC50 between paired and non-paired voles revealed that a lower dose of the KOR agonist was needed to achieve a 50% decrease in stimulated DA release within the NAc shell of pair bonded males (t-test; IC50; $t_{(6)} = 4.92$, p=0.002, *Figure 7h*). The slope of the concentration response curve also significantly differed between paired and non-paired males (t-test; $t_{(6)} = 3.74$, p=0.009, *Figure 7i*). Given that the density of KOR binding is reduced in pair bonded males, these data suggest that pair bonding may result in mechanistic changes in the function of the KOR in males, but not females, as similar measures did not significantly differ between paired and non-paired females (IC50: t-test; $t_{(6)} = 1.36$, p=0.22; *Figure 7J* and slope: t-test; $t_{(6)} = 0.55$, p=0.60; *Figure 7k*). Moreover, direct comparisons of the concentration response curves for paired males and females identified a leftward shift in the concentration curve of paired males (two-way ANOVA, $F_{(1, 7)} = 9.60$, p=0.02) and multiple doses that produced significantly greater inhibition of DA release in paired males compared to females (Bonferroni's post hoc test; 0.1 μM, p=0.02; 0.3 μM, p=0.004; 1 μM, p=0.001; 3 μM, p=0.009) (*Figure 7—figure supplement 2*). However, it should be noted that paired males and females did not significantly differ in IC50 or slope of the concentration response curve (*Figure 7—figure supplement 2*). Nonetheless, the present data suggest that although KOR binding is reduced within the NAc shell of pair bonded males, the function of these receptors may be enhanced as greater reductions in KOR induced decreases in stimulated DA release occurred within the NAc shell of pair bonded males.

In summary, data from the present study reveal that both the DA and dynorphin/KOR systems within the NAc shell undergo sex-specific alterations following the establishment of a pair bond (*Figure 8a,b*). Specifically, both sexes show increases in D1 receptor and dynorphin mRNA within

**Table 4.** Confidence intervals for IC50's in KOR agonist dose response study.

| Group | 95% Confidence interval |
| --- | --- |
| Male non-pair bonded | 4.761818 to 13.906848 |
| Male pair bonded | −1.2093417 to 4.4570177 |
| Female non-pair bonded | 0.73751 to 18.11349 |
| Female pair bonded | −0.73394 to 10.70294 |

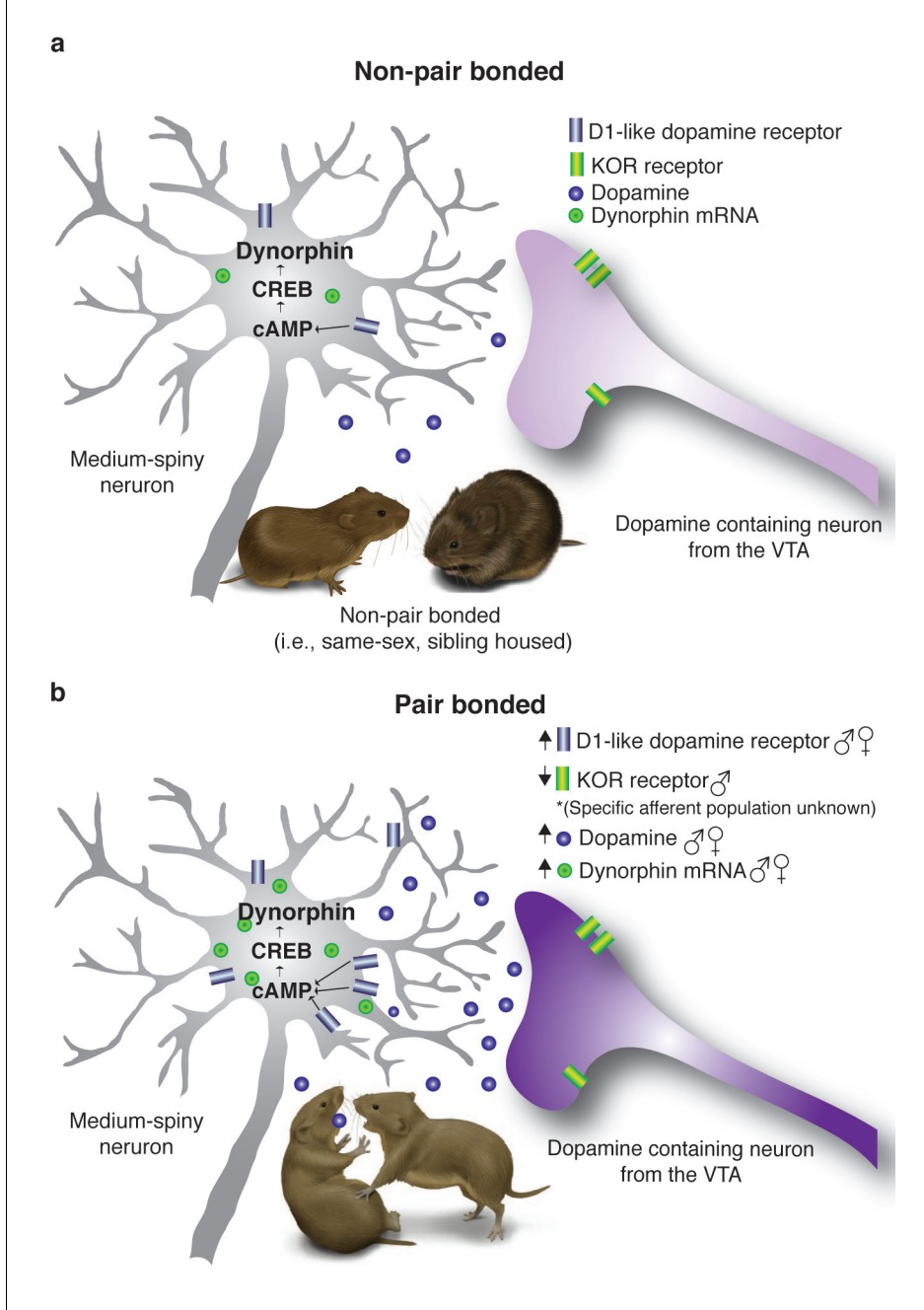

**Figure 8.** Pair bonding alters DA and dynorphin/KOR systems within the ventral striatum. (**a**) Non-pair bonded prairie voles readily approach novel conspecifics and have lower levels of stimulated DA release as well as *Drd1* and *Pdyn* mRNA expression within the ventral striatum. (**b**) Following the establishment of a pair bond, male and female prairie voles aggressively reject novel conspecifics and the ventral striatum undergoes a dramatic reorganization. Specifically, pair bonding enhances DA release within the NAc shell as well as up-regulates *Drd1* as well as *Pdyn* within the ventral striatum of both males and females. Pair bonded males also show an additional decrease in KOR binding within the NAc shell.

the ventral striatum as well as enhanced DA transmission within the NAc shell. However, in relation to the dynorphin/KOR system, only males showed an overall reduction in membrane expression of KORs as well as dramatic reductions in DA transmission in response to a KOR agonist. Given that these systems are known to directly interact with each other (***Engber et al., 1992***; ***Carlezon et al.,***

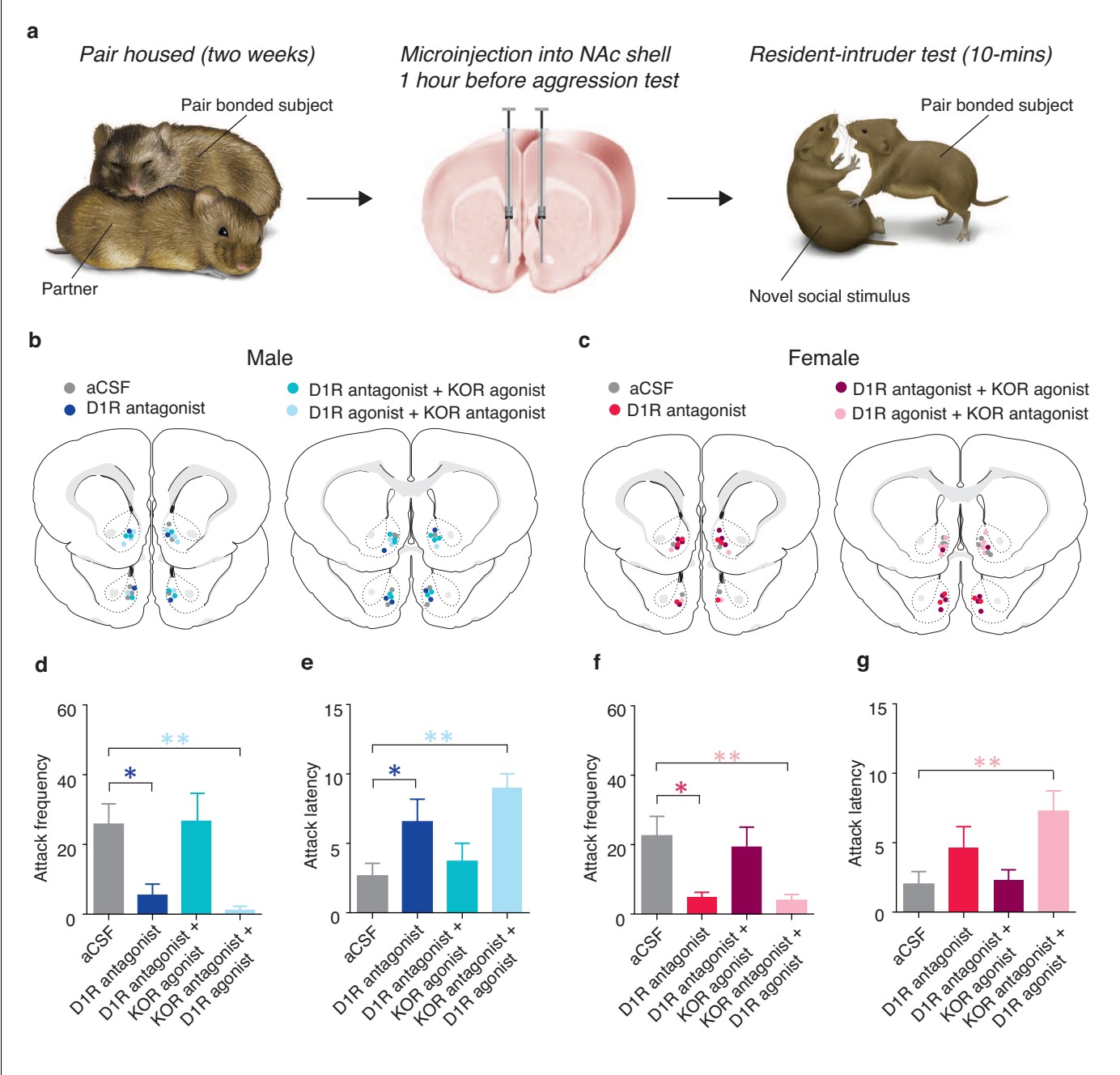

**Figure 9.** Interactions between D1-like and KORs mediate pair bond maintenance. (a) Experimental Design. (b,c) Histological location of injection sites in (b) males and (c) females. (d) Compared to control pair-bonded males that received site-specific infusions of aCSF prior to resident-intruder testing (n = 6), males that received site-specific infusions of a D1-like receptor antagonist into the NAc shell showed attenuated levels of selective aggression as well as (e) increased attack latency toward intruders (n = 6). However, aggression levels and attack latencies were returned to normal when the antagonist for the D1-like receptor was administered in combination with a KOR agonist (n = 7) suggesting that D1-mediated aggression occurs through downstream activation of KORs. This interaction was confirmed by the ability of the KOR antagonist to attenuate selective aggression even when it was administered in combination with the D1-like receptor agonist (n = 7). (f,g) Similar to males, blockade of D1-like receptors within the NAc shell of paired females (n = 6) attenuated selective aggression compared to aCSF controls (n = 6). Aggression frequency was returned to the level of paired female controls when the D1-like receptor antagonist was administered in combination with a KOR agonist (n = 7). Finally, the attenuation of attack frequency and the increase in attack latency mediated by a KOR antagonist was maintained even in the presence a D1-like receptor agonist (n = 6). Summary data are presented as mean ± SEM. *p<0.05, **p<0.005.

*1998*; *Ebner et al., 2010*; *Chartoff et al., 2016*) and that activation of both D1-like receptors (*Aragona et al., 2006*) and KORs are required for the expression of selective aggression (*Resendez et al., 2012*), we next determined if these systems interact in vivo to regulate pair bond maintenance.

## D1-like and KORs interact to mediate selective aggression

Previous studies have shown that the DA and the dynorphin/KOR systems function in sequence of each other, with stimulation of D1-like receptor promoting downstream activation of the dynorphin/KOR system (*Gerfen et al., 1990*; *Carlezon et al., 1998*). We therefore tested the hypothesis that D1-like receptor regulation of selection aggression is upstream of its regulation by KORs. Similar to the anatomical characterization studies described above, prairie voles were paired with an opposite sex conspecific for 2 weeks to allow sufficient time for a pair bond to be established. At the end of the cohabitation period, site-specific behavioral pharmacology was utilized in combination with resident intruder testing to examine the sequential nature of interactions between activation of D1-like receptors and KORs on the expression of selective aggression (*Figure 9a*). More specifically, if KOR activation is indeed downstream of D1-like receptor activation than activation of KORs despite pharmacological blockade of D1-like receptors should still result in the expression of selective aggression. Conversely, pharmacological manipulations that would result in a reduction in KOR activation, such as administration of a D1-like antagonist in the absence of a KOR agonist or administration of a KOR antagonist in the presence of a D1-like receptor agonist, should attenuate the expression of selective aggression.

Compared to control subjects receiving site specific administration of aCSF, pharmacological manipulation of NAc shell D1-like and KORs (*Figure 9b,c*) significantly altered the expression of selective aggression in both pair bonded males (Attack frequency: one-way ANOVA, $F_{(3,25)}$ = 5.55, p=0.005, *Figure 9d*; Attack latency: one-way ANOVA, $F_{(3,25)}$ = 5.54, p=0.005, *Figure 9e*) and females (Attack frequency: one-way ANOVA, $F_{(3,23)}$ = 4.59, p=0.01, *Figure 9f*). However, attenuation of selective aggression was dependent on the combination of agonists and antagonists administered. As expected, pharmacological blockade of NAc shell D1-like receptors significantly attenuated measures of selective aggression in both pair bonded males (Attack frequency: planned contrast, post hoc: p=0.03, Attack latency: planned contrast, post hoc: p=0.04) and females (Attack frequency: planned contrast, post hoc: p=0.01). However, blockade of NAc D1-like receptors did not significantly attenuate attack latency in females (one-way ANOVA, $F_{(3,23)}$ = 4.77, p=0.01, p=0.47). Thus, activation of NAc shell D1-like receptors is required for the expression of selective aggression in both sexes, possibly due to D1-like receptor mediated activation of the dynorphin/KOR system.

To determine if KOR regulation of selective aggression is indeed downstream of the DA system, we co-administered a KOR agonist along with a D1-like receptor antagonist, resulting in KORs to be activated despite the inhibition of D1-like receptors. Activation of KORs in the presence of the D1-antagonist restored selective aggression as mean attack frequency (Male: planned contrast, post hoc: p=0.92; Female: planned contrast, post hoc: p=0.62) and attack latency (Male: planned contrast, post hoc: p=0.54; Female: planned contrast, post hoc: p=1.00) did not differ from paired controls in either sex, suggesting that D1-like receptors mediate selective aggression through downstream activation of the dynorphin/KOR system. In contrast, D1-like receptor activation in the presence of KOR inactivation was insufficient to restore measures of selective aggression to levels of paired controls (Male attack frequency: planned contrast, post hoc: p=0.006; Female attack frequency: planned contrast, post hoc: p=0.008; Male attack latency: planned contrast, post hoc: p=0.001; Female attack latency; planned contrast, post hoc: p=0.005; *Figure 9d–g*), further suggesting that KOR mediation of selective aggression is downstream of D1-like receptors. Finally, these manipulations specifically altered aversively motivated behaviors as there were no differences in affiliative (Male: one-way ANOVA, $F_{(3,25)}$ = 1.95, p=0.15; Female: one-way ANOVA, $F_{(3,23)}$ = 1.58, p=0.23) or locomotor behavior (Male: one-way ANOVA, $F_{(3,23)}$ = 0.75, p=0.54; Female: one-way ANOVA, $F_{(3,23)}$ = 0.69, p=0.57) (data not shown). Together, these data support the hypothesized mechanism that DA activation of D1-like receptors promotes downstream release of dynorphin to subsequently activate KORs within the NAc shell and generate selective aggression.

In addition to regulation of selective aggression, D1-like receptors within the NAc shell have also been shown to mediate the protective effects of pair bonding against drug reward (*Liu et al., 2011*).

Given the identification that D1-like receptor regulation over pair bonding occurs through downstream activation of the dynorphin/KOR system, it is also possible that the protective effects of pair bonding are mediated though activation of this aversive processing system. We therefore next determined if activation of the dynorphin/KOR system is required for pair bonding to exert protective effects against the rewarding properties of amphetamine (AMPH).

## Amphetamine-induced neuroplasticity mimics that of pair bonding

Positive social relationships, such as the formation of strong social ties, modify the brain in such a manner that results in an attenuation of the rewarding properties of drugs of abuse (*Creswell et al., 2015*). Thus, identifying overlapping neural systems that mediate both social bonding and drug reward processing may have positive therapeutic value in the treatment of addiction. We therefore first examined the impact of a rewarding regimen of AMPH (3 AMPH injections at 1-mg/kg across 3 days) on KOR binding in non-pair bonded prairie voles. This dose of AMPH was chosen because it is well established to elicit a preference for AMPH in the conditioned place preference task in both male and female prairie voles (*Aragona et al., 2007*; *Liu et al., 2010*; *2011*).

Compared to control males (3 injections of saline across 3 days), male subjects exposed to a rewarding regimen of AMPH showed significantly altered patterns of striatal KOR expression (two-way ANOVA, $F_{(1,78)}$ = 15.97, p=0.0001, *Figure 10a,b*). Moreover, the pattern of AMPH induced alterations in KOR expression within the striatum of males was similar to the pattern induced by pair bonding, with AMPH exposure significantly reducing KOR expression within the dorso-medial (Bonferroni's post hoc test, p=0.02) and ventral NAc shell (Bonferroni's post hoc test, p=0.01)

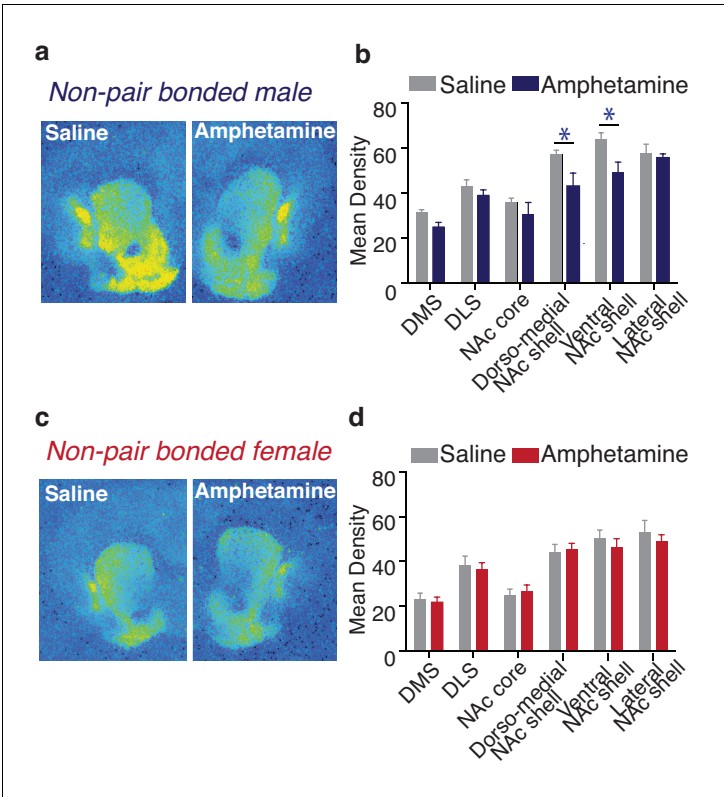

**Figure 10.** Amphetamine decreases KOR binding within the striatum of males. (a,b) AMPH decreased KOR binding within the dorso-medial and ventral NAc shell of non-pair bonded males (n = 7–8/group). (c,d) Similar to pair bonding, AMPH did not impact striatal KOR binding in females (n = 7–8/group). Summary data are presented as mean ± SEM. *p<0.05.

The following figure supplement is available for figure 10:

**Figure supplement 1.** Sex differences in prairie vole KOR binding density following amphetamine exposure.

**Table 5.** Non-significant statistics for comparisons of KOR binding density in saline versus amphetamine treated males.

| Striatal sub-region | Bonferonni's *post hoc* test |
|---|---|
| Dorso-medial striatum | p>0.99 |
| Dorso-lateral striatum | p>0.99 |
| NAc core | p>0.99 |
| Dorso-lateral striatum | p>0.99 |

(*Figure 10b*, *Table 5*). In contrast, AMPH exposure had no significant impact on striatal KOR expression in females (two-way ANOVA, $F_{(1,108)}$ = 0.44, p=0.51, *Figure 10c,d*). Moreover, when the pattern of KOR expression binding was directly compared between males and females (two-way ANOVA, $F_{(1,96)}$ = 39.80, p=0.0001), control subjects significantly differed in KOR binding density with control males having significantly higher levels in the dorso-medial (Bonferroni's post hoc test, p=0.03) and ventral NAc shell (p=0.0002) (*Figure 10—figure supplement 1*). However, as with the experience of pair bonding, AMPH exposure also eliminated these sex differences (two-way ANOVA, $F_{(1,90)}$ = 1.89, p=0.17) (*Figure 10—figure supplement 1*). Given that AMPH altered male, but not female, striatal KOR binding density, we focused next set of experiments on male subjects.

## Social reward impairs AMPH-induced place conditioning

Pair bonding exerts protective effects against AMPH reward (*Liu et al., 2011*); however, the establishment of a pair bond is associated with a complex suite of socially related experiences, such as exposure to a novel social stimulus, extended periods of cohabitation, the development of social familiarity, copulation, and impregnation and it is not well understood how these individual components contribute to the neural protective effects that pair bonding exerts against drug reward (*Resendez et al., 2013*). To this end, we conducted a detailed analysis of pair bond associated social experiences that may contribute to the social buffering of drug reward. Specifically, male subjects were randomly assigned to one of the following treatment groups: social familiarity (i.e., same-sex sibling housed), extended cohabitation with a novel social stimulus without mating (i.e., 2 weeks cohabitation with a novel male or ovariectomized female), or extended cohabitation with a reproductive partner (2 weeks cohabitation with an intact female). Given that not all gonadally intact male-female pairs achieved pregnancy, males housed with an intact female were further categorized by the reproductive status of the female partner at the completion of testing (i.e., no indication of pregnancy, sub-optimally pregnant, or optimally pregnant). Exposing males to these different social experiences as well as categorizing mating pairs by their reproductive status allowed us to determine the influence of each social condition on the protective effects of pair bonding (*Figure 11a*).

Following exposure to one of the above described social conditions, males underwent conditioned place preference procedures to identify the specific aspects of male pair bonding that contribute to the attenuation of AMPH reward. A separate group of same-sex sibling housed males was conditioned with saline only and the duration of time spent in the AMPH paired chamber during the post-test session was compared to this treatment group. Social experiences that do not result in the establishment of a pair bond failed to protect against AMPH reward (one-way ANOVA, $F_{(4,69)}$ = 0.67, p=0.0001) as male subjects paired with a same-sex sibling (Tukey's post hoc test, p<0.0001), novel male (p=0.004), or OVX female (p=0.008) formed significant preferences for the AMPH paired chamber (*Figure 11a*). In contrast, males housed under conditions that promote pair bonding did not form a preference for the AMPH paired chamber (p=0.14). Moreover, when these males were further classified by the pair's pregnancy status, only males from pregnant pairs exhibited protection against AMPH reward as males from both optimally (p>0.99) and sub-optimally pregnant pairs (p=0.94) did not form a preference for the AMPH paired chamber, while males from non-pregnant pairs formed significant preferences (p=0.006) (*Figure 11a* inset). Together, these data indicate that the establishment of a fully developed pair bond, and not the other associated social experiences, mediates social buffering of drug reward.

To further explore the influence of pair bonding on drug reward, we examined the relationship between pair fecundity and preference for the drug-paired chamber. While males from both sub-

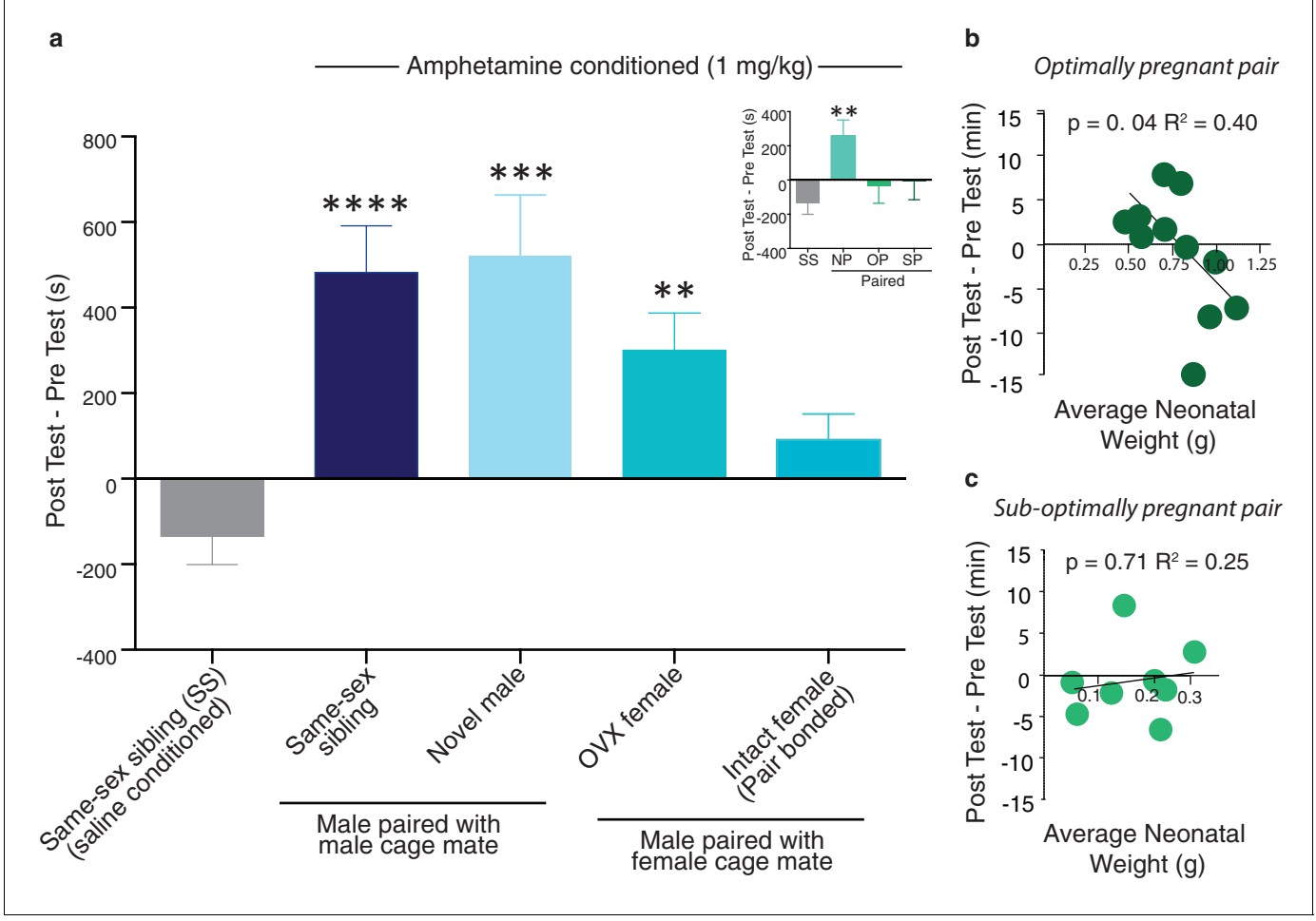

**Figure 11.** Neural protection against drug reward is specific to pair bonding in males. (**a**) Male prairie voles were housed with a familiar cage mate, a novel male, an ovariectomized (OVX) female, or an intact female for two weeks prior to AMPH conditioning. Compared to saline treated males, all groups except males housed with an intact female formed a preference for the AMPH paired chamber (n = 6–33/group). To determine if pregnancy status influenced the rewarding properties of AMPH, males housed with an intact female were further classified by the pairs pregnancy status (inset). Only males paired with a female that became pregnant (suboptimally (SP) or optimally (OP)) during the 2-week pairing period were protected against the rewarding properties of amphetamine as males paired with females that were not pregnant (NP) formed a preference for the AMPH paired chamber. (**b**) The establishment of an optimal pregnancy strongly influenced the rewarding properties of AMPH as there was a negative correlation between the duration of time spent in the AMPH paired chamber and the gestational stage of the female for optimally pregnant pairs (n = 11). (**c**) In contrast, there was no relationship between pregnancy stage and AMPH preference for sub-optimally pregnant pairs (n = 8). Summary data are presented as mean ± SEM. *p<0.05, **p<0.005, ***p<0.0005.

optimally and optimally pregnant pairs showed some degree of protection against AMPH reward, males from optimally pregnant pairs showed the strongest relationship between fecundity and the attenuation of AMPH reward. Specifically, in males from optimally pregnant pairs, the rewarding properties of AMPH were negatively correlated with the pregnancy status of the female (linear regression, $R^2$ = 0.403, $F_{(1,9)}$ = 6.079, p=0.036, *Figure 11b*). However, a similar relationship was not found in males from sub-optimally pregnant pairs (linear regression, $R^2$ = 0.025, $F_{(1,6)}$ = 0.155, p=0.708, *Figure 11c*). Thus, the reproductive status of the pair influences pair bond induced protection against AMPH reward.

## NAc shell KORs attenuate the rewarding properties of amphetamine in pair bonded males

We next determined if pair bond induced alterations in the male prairie vole KOR system contribute to neural protective effects against AMPH reward. Similar to above, males were paired with either a

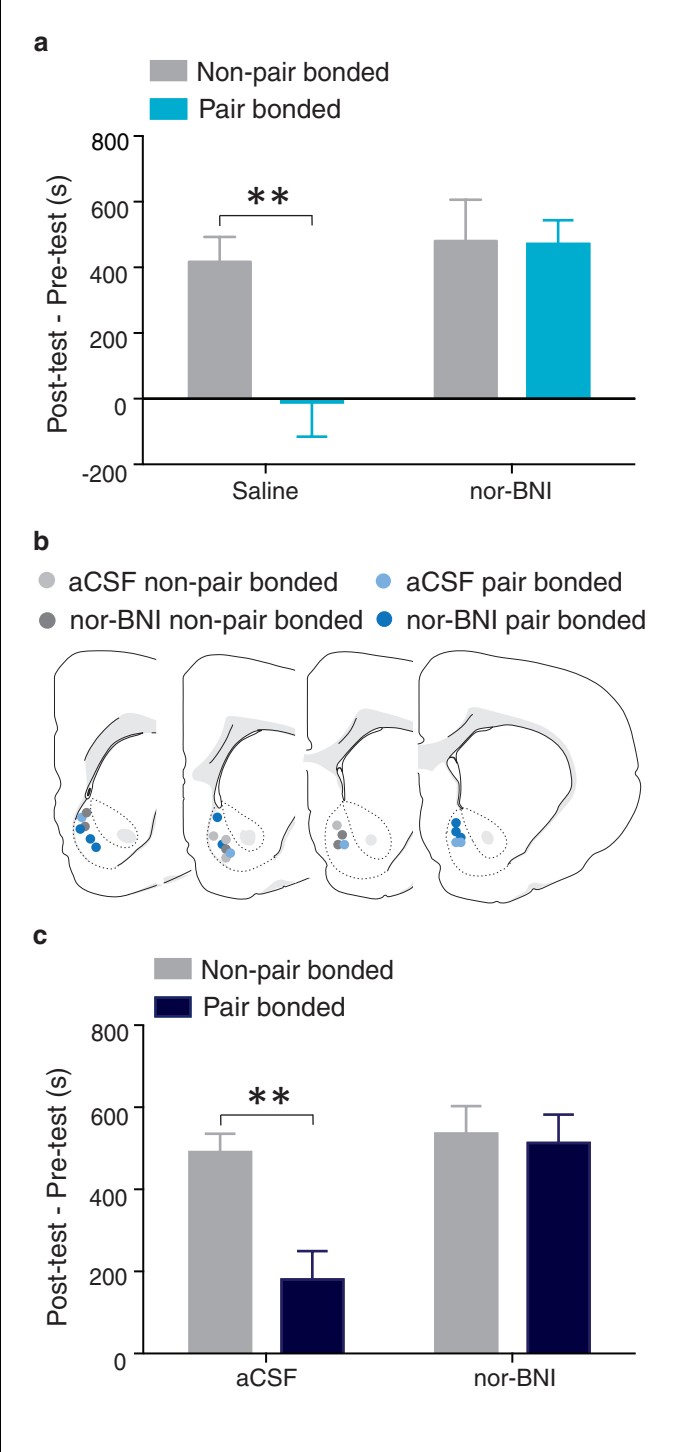

**Figure 12.** NAc shell KORs mediate the protective effects of pair bonding. (**a**) Peripheral administration of nor-BNI restored the rewarding properties of AMPH for paired males (n = 6–14/group). (**b**) Histological location of injection sites. (**c**) Site-specific blockade of NAc shell KORs was sufficient to alleviate pair bond induced attenuation of AMPH reward (n = 4–7/group). Summary data are presented as mean ± SEM. **p<0.005.

male partner (non-paired) or an intact female (paired) for 2 weeks prior to AMPH conditioning. On day one of conditioning, males in both groups received either peripheral administration of saline or a KOR antagonist. AMPH preference varied by housing conditioning as well as treatment (two-way

ANOVA, $F_{(1, 28)}$ = 8.13, p=0.008). Specifically, compared to non-paired males that received saline injections prior to AMPH conditioning, paired males treated with saline spent significantly less time in the AMPH paired chamber (Bonferroni's post hoc test, p=0.004, *Figure 12a*). In contrast, global blockade of KORs in pair bonded males restored the rewarding properties of AMPH as paired males that received peripheral injections of nor-BNI prior to AMPH conditioning did not differ from non-paired males in the duration of time spent in the AMPH paired chamber (p>0.99, *Figure 12a*). Together, these data indicate that the protective effects of pair bonding are in part mediated by KORs.

To determine if KOR buffering of AMPH reward is mediated within the NAc shell, we next examined if blockade of KORs specifically within this region was sufficient to restore AMPH preference in pair bonded males (*Figure 12b*). A two-way ANOVA indicated that AMPH preference varied by housing condition and treatment ($F_{(1, 29)}$ = 8.33, p = 0.007, *Figure 12c*). Compared to non-paired males, paired males that received site-specific injections of aCSF into the NAc prior to AMPH conditioning spent significantly less time in the AMPH paired chamber (Bonferroni's post hoc test, p = 0.003, *Figure 12c*). However, pair bonded males that were administered nor-BNI prior to AMPH conditioning did not differ from non-paired males that also received site-specific administration of nor-BNI in the duration of time spent in the AMPH paired chamber (p>0.99, *Figure 12c*). Together, these data demonstrate that KORs in the NAc shell are indeed involved in pair bond induced neuroprotection against drug reward as activation of these receptors is required for pair bond induced attenuation of drug reward.

## Discussion

Prior to forming a pair bond, sexually naïve prairie voles perceive novel social stimuli as rewarding. However, following extended cohabitation with a breeding partner and the full development of a pair bond, prairie voles aggressively reject novel conspecifics. In the present study, we critically extend current understanding of the neural mechanisms that mediate the transition to the pair bonded state by identifying neuroplasticity within the NAc shell that regulates the maintenance of monogamous bonds. Specifically, compared to non-paired subjects, pair bonded males and females have enhanced DA release potential within the NAc as well as an up-regulation of D1 receptor and dynorphin mRNA within this region. The functional significance of this pair bond induced plasticity was demonstrated with site-specific pharmacology as blockade of either one of these receptor systems attenuates the expression of pair bond maintenance. Moreover, while both sexes show some degree of mate guarding behavior, paired males are generally more aggressive than paired females and in the present study we provide the first mechanistic data to explain sex differences in the expression of selective aggression. Compared to females, pair bonded males exhibited additional neuroplasticity within the NAc as well as fecundity dependent alterations in DA transmission including: a positive correlation between DA release and pair fecundity, decreased binding of KORs, and enhanced KOR regulation over DA transmission. These more robust alterations within the NAc of males may partially explain why males show more intense mate guarding than females as well as the dependence of male mate guarding on fecundity. Finally, we also show that AMPH exposure alters KOR binding density within a sex-specific manner and that neuroplasticity within the dynorphin/KOR system of pair bonded males is critical for pair bond induced attenuation of drug reward. Together, the data presented in the present study reveal novel mechanisms underlying the maintenance of monogamous pair bonds as well as the neural protective effects of social bonding against drug reward.

### Neural and behavioral plasticity associated with pair bonding

A combination of comparative anatomical approaches and behavioral pharmacological studies has been utilized to identify neural mechanisms that underlie pair bond formation and maintenance in the socially monogamous prairie vole (*Carter et al., 1997*; *Bales et al., 2007*; *Young et al., 2008*; *Aragona et al., 2009*). In regards to pair bond formation, the initial stages of bond development are mediated in part by D2-like, oxytocin, vassporessin, and mu-opioid receptors systems located within reward processing regions of the brain such as the striatum and the ventral pallidum (*Insel et al., 1998*; *Young et al., 2008*; *Resendez and Aragona, 2013*). Interesting, following social conditioning procedures that promote the establishment of a pair bond, we did not identify

alterations in the expression of mRNA for genes encoding D2-like, oxytocin, vasopressin, or mu-opioid receptors (*Carter et al., 1997*; *Johnson and Young, 2015*). One possible explanation of why a lack of an alteration in receptor systems associated with partner preference behavior was not observed in the present study is that a preference for social proximity is required in both the bonded and non-bonded state, while the emergence of an aversion towards social novelty is specific to the establishment of a pair bond and requires alterations in both social and motivational circuitry (*Resendez and Aragona, 2013*).

In contrast to neural systems that regulate pair bond formation, motivational and valence processing systems associated with pair bond maintenance undergo a dramatic overhaul during the transition from the naive to the pair bonded state. An important function of this alteration is to render pair bonded voles hyper aggressive toward novel conspecifics in order to achieve robust mate guarding (*Resendez and Aragona, 2013*). As such, it is not surprising that neuroplasticity associated with pair bond development occurs within systems that mediate the expression of social behaviors associated with pair bond maintenance, DA/D1 and dynorphin/KOR systems and that these alterations occur specifically within the NAc shell, the striatal sub-region where these receptor systems act to mediate pair bond maintenance. Moreover, behavioral pharmacological data make evident the functional significance of these alterations by demonstrating that activation of KORs within the NAc shell of prairie voles mediates the assignment of negative social valence onto novel conspecifics, while blockade of either D1-like or KORs abolishes selective aggression. Together, these data support the working hypothesis that neuroplasticity within the NAc shell stabilizes established pair bonds by enhancing the perceived negative valence of novel social stimuli and the promotion of robust mate guarding.

## Sex-specific alterations in motivational systems and behavior

In species where monogamous breeding systems have evolved, males will often engage in mate guarding behavior to prevent access of competitor males to the female partner, increasing the males opportunity for selective breeding as well as reducing the likelihood of uncertain paternity (*Trivers and Campbell, 1972*; *Kleiman, 1977*). While this breeding strategy is adaptive for some species, mate guarding behavior is intensely energetically costly given that a great deal of time must be spent maintaining proximity to the female and a high level of energy expended engaging in risky agonistic social encounters with competitor males (*Parker, 1974*; *Grafen and Ridley, 1983*; *Getz et al., 1990*; *Crews and Moore, 1986*). As a result, males can usually only successfully guard one female at a time (*Brotherton et al., 2003*) and, to maximize reproductive fitness, it is adaptive for mate guarding behavior to emerge only after the establishment of a reproductively successful pair bond (*Resendez et al., 2012*). For the socially monogamous prairie vole, an indication of pair fecundity is indeed required for males to transition to the pair bonded state (*Curtis, 2010*; *Resendez et al., 2012*); yet the neural mechanisms that underlie this sex difference have remained elusive. In the present study, we provided the first proximal mechanistic data to explain how fecundity induced sex differences in selective aggression might arise.

Consistent with the theory that mate guarding in monogamous males serves to maximize reproductive fitness (*Trivers and Campbell, 1972*), the present study demonstrates that the extent to which DA transmission was enhanced within the NAc shell of paired males was positively correlated with both pair fecundity as well as measures of selective aggression. Moreover, significant enhancements in NAc shell DA transmission only occurred in males from optimally pregnant pairs and males from these pairs also showed significantly higher levels of selective aggression. Thus, neuroplastic changes that are associated with the expression of selective aggression only occur in paired males when an adaptive benefit in defending the female partner has been established (i.e., when the benefit of defending the female outweighs the risk associated with aggressive conflict as well as predation risks that may occur when searching for a new mate).

In contrast to paired males, a relationship between alterations in DA transmission and pair fecundity were not identified in paired females, which is consistent with both field (*Rose and Gaines, 1976*) and laboratory (*Resendez et al., 2012*) studies that have yet to identify a relationship between fecundity and mate guarding behavior in female prairie voles. One possible explanation for a lack of influence for fecundity on both neural and behavioral changes associated with pair bonding in females is a difference between males and females in the ultimate mechanisms that underlie the decision to bond with a partner. For instance, given that the reproductive status of the male is

constant, whereas females require extended periods of contact with a male for sexual receptivity to be induced (*Carter et al., 1987*), it may be more beneficial for a female to increase the likelihood of reproductive success by remaining in contact with a male partner than to risk predation by searching for a new mate. Moreover, unlike males, female mammals do not risk exerting unnecessary energy raising offspring that are not their own if their male partner engages in extra-pair copulations (*Trivers and Campbell, 1972*). Thus, the risk to reward ratio for engaging in agonistic social encounters may not be as great for females and may also partially explain why females tend to be less aggressive overall than males. Together, sex differences in the adaptive value of mate guarding as well as sex differences in temporal and environmental factors that regulate reproductive activation between males and females may partially underlie known sex differences in the intensity of selective aggression as well as behavioral and neural neuroplasticity associated with the transition to the bonded state.

## Sub-region specific alterations in male NAc shell KOR binding

Increasing evidence suggests that anatomical and functional heterogeneity occurs within the shell region of the NAc (*Peciña and Berridge, 2005*; *Resendez and Aragona, 2013*; *Richard et al., 2013*). More recently, it has been demonstrated that heterogeneity in the valence coding properties of the dynorphin/KOR system is anatomically segregated within the NAc shell (*Castro and Berridge, 2014*; *Al-Hasani et al., 2015*). Specifically, the release of dynorphin into the dorso-medial region of NAc shell and the subsequent activation of KOR induces positive hedonics (*Castro and Berridge, 2014*; *Al-Hasani et al., 2015*), while the release of dynorphin into the ventral NAc shell induces aversion (*Al-Hasani et al., 2015*). Interestingly, pair bonding altered the binding density of male NAc shell KORs in a sub-region specific manner that maps onto the topographical organizational of the aversive coding properties of the dynorphin/KOR system. Pair bonding down-regulated KOR binding within the ventral region of the NAc shell, while leaving KOR binding within the dorso-medial and lateral regions of the NAc shell unaltered. Thus, pair bonding altered KOR binding only in the sub-region of the NAc shell where activation of these receptors acts to encode aversion.

One possible mechanism underlying the sub-region specific influence of the dynorphin/KOR system on valence coding may be due to anatomical heterogeneity in the downstream projection targets of the dorso-medial and ventral regions of the NAc shell. In general, downstream projection targets from the afferents of cell bodies originating in the dorsomedial region of the NAc shell are more widespread than those originating from cell bodies located in the ventral region of the NAc shell. For cell bodies located in the dorso-medial NAc shell, the highest density of afferents are located in the medial region of the ventral pallidum (VP), the lateral preoptic area, and the lateral hypothalamus with sparser fiber labeling occurring within the lateral septum, the bed nucleus of the stria terminalis, the anterior hypothalamus, the medial preoptic area, and rostral portion of the ventral tegmental area (*Thompson and Swanson, 2010*; *Zahm et al., 2013*). In contrast, the ventral region of the NAc shell varies from the dorso-medial region of the NAc shell in both the number and specific brain regions it projects to. While the dorso-medial NAc shell sends dense projections to the medial region of the VP (*Thompson and Swanson, 2010*), the ventral region of the NAc shell projects specifically to the lateral region of the VP and also sends sparser projections to the lateral preoptic area and rostral-caudal extent of the ventral tegmental area (*Zahm et al., 2013*). It is therefore possible that although the release of dynorphin and the subsequent activation of NAc shell KORs has the potential to reduce dopaminergic and glutamatergic transmission throughout the entire dorsal-ventral axis of the NAc shell (*Hjelmstad and Fields, 2001*; *Britt and McGehee, 2008*), the sum of the influence on downstream neuronal networks has the potential to vary greatly between the two sub-regions.

One notable downstream projection target that is unique to the ventral NAc shell is the lateral region of the VP. The lateral VP is an important brain region for reward processing (*Cromwell and Berridge, 1993*) and is innervated by NAc shell medium spiny neurons that primarily express Gi-coupled D2-like DA receptors (*Gerfen and Young, 1988*). Thus, KOR-mediated reductions in DA within the ventral NAc shell would increase the activity of GABAergic medium spiny neurons that project to the lateral VP, resulting in the inhibition of this region (*Bonci and Carlezon, 2005*). Interestingly, reduced activity has been observed directly within the VP following exposure to aversive stimuli (*Itoga et al., 2016*) as well as an increase in activity within brain nuclei that receive input specifically from GABAergic neurons located within the lateral region of VP. Of specific interest is the relief of

inhibition of the periventricular nucleus of the thalamus (PVT), a brain nucleus that is downstream of the VP and has been indicated in aversive processing (*Yashoshima et al., 2007*). In contrast to the ventral NAc shell, reductions in DA specifically within the dorso-medial region of the NAc shell would cause a decrease in the activity of the PVT through inhibition of glutamatergic afferents from the LH that project to the PVT (*Thompson and Swanson, 2010*; *Zahm et al., 2013*). Thus, the ventral NAc shell → lateral VP → PVT circuit may be one mechanism in which site-specific modulation of NAc shell KORs contributes to aversive coding, while the dorso-medial NAc shell → LH → PVT circuit may contribute to positive valence coding by NAc KORs. However, more work is necessary to determine how topographical organization of NAc shell contributes to opposing modulation of valence coding and the subsequent divergent responses on motivated behavioral states, such as approach versus avoidance behaviors.

## Interactions between D1-like and KORs mediate pair bond maintenance

Complex behaviors often require interactions between multiple neural systems. Indeed, studies of pair bond formation show that partner preferences require concurrent activation of D2-like and OT receptors within the NAc shell (*Liu and Wang, 2003*) as well as V1a receptor activation within the VP (*Lim et al., 2004*). These previous studies argued that peptide systems are necessary for social recognition, while DA transmission is important for reward processing (*Young and Wang, 2004*; *Johnson and Young, 2015*). In the present study, we expand our current understanding of neural mechanisms involved in the regulation of pair bond behavior by demonstrating that interactions between opioid peptides, such as the dynorphin/KOR system, and DA transmission within the NAc may act to couple valence processing systems within motivational circuitry. Consistent with the theory that interactions between these systems mediate pair bond induced transitions in the valence encoding of novel social stimuli, previous studies of drug reward have demonstrated that interactions between DA/D1 and dynorphin/KOR systems mediate the propensity for previously rewarding stimuli to be processed as aversive. Specifically, stimulation of D1-like receptors phosphorylates cAMP response element binding protein (CREB) to induce the expression and release of dynorphin (*Carlezon et al., 1998*), resulting in the valence encoding of a psychostimulant to be reversed from rewarding to aversive (*Pliakas et al., 2001*). Together, these data suggest that alterations in the activity and, subsequently, interactions between DA and dynorphin/KOR systems play a critical role in experience mediated plasticity in reward processing.

While it is known that activity within the dynorphin/KOR system plays a critical role in reward processing, the mechanism in which this system modulates the encoding of reward is not well understood. One hypothesized mechanism in which the dynorphin/KOR is thought to negatively impact motivation and reward processing is through its ability to robustly decrease dopaminergic transmission within the NAc (*Shippenberg et al., 1996*; *Carlezon and Thomas, 2009*). Interestingly, interactions between these systems were augmented following the establishment of a pair bond in male, but not female, prairie voles. Given that paired bonded males are the more aggressive sex and that paired males also incur greater reproductive costs if their mate engages in extra-pair copulations (*Resendez et al., 2012*), it is possible that paired males show a greater aversion to social novelty and this enhanced aversion may be mediated by augmented coupling between DA and dynorphin/KORs within the NAc. While more work is necessary to determine the exact mechanism in which NAc KORs mediate the expression of sex differences in selective aggression, the present study nonetheless provides an interesting example of how the dynorphin/KOR system modulates reward and motivation to promote ethologically relevant behavioral adaptation.

## Neuroprotective effects of pair bonding on drug reward

Addiction is a debilitating disorder that is characterized in part by chronic relapse, and, while pharmacological treatments continue to be sought for the treatment of this disorder, many have been ineffective in sustaining drug abstinence (*Fattore and Diana, 2016*). Interestingly, there is compelling evidence that a preventative approach, focused on neural adaptations resulting from the formation and maintenance of positive social relationships, may offer positive benefits to psychological well being (*Feldman, 2015*). For example, in drug-addicted humans, the presence of positive social support reduces the propensity for relapse (*Creswell et al., 2015*), suggesting that positive social experience, such as bonding, may wire the brain in a manner that reduces future drug seeking

behavior (*Young et al., 2011*). Yet, despite the demonstrated positive benefit for social bonding on mental health, the neural mechanisms underlying the relationship between social bonding and drug taking have not been extensively studied. The scarcity of studies examining this relationship is likely related to a lack of animal models that exhibit both the propensity for social bonding and drug taking behavior.

The socially monogamous prairie vole offers an excellent animal model in which to study the relationship between drug and social reward because, unlike most mammals, they form selective social attachments to their mating partner. Importantly, as demonstrated in the present study, the establishment of a pair bond, but not mere social exposure or mating, is required for social bonding to exert protective effects against AMPH reward. The specificity of the establishment of a social bond to the neural protection against drug reward is likely related to the fact that the establishment of a pair bond alters the brain in a manner in which other social experiences do not (*Liu et al., 2011*; *Smith and Wang, 2014*). Indeed, it has previously been demonstrated that activation of neural systems that mediate pair bond maintenance (NAc D1-like receptors), but not those that mediate pair bond formation (NAc D2-like receptors), are required for pair bond induced protection against drug reward (*Liu et al., 2011*). Here, we extended these previous findings by demonstrating that, for male prairie voles, activation of NAc shell KORs is required for pair bond induced attenuation of AMPH reward. Given that drugs that act at the dynorphin/KOR system are currently under intense investigation as potential therapeutics for the treatment of addiction (*Carlezon and Miczek, 2010*) and that pair bonding induces neural plasticity within this system to result in an attenuation in drug reward, there may be considerable therapeutic value in understanding the neural mechanisms in which social experiences alter reward circuitry to buffer against drug reward. Finally, data presented in the present study also provide support for the consideration of socially related cognitive therapies when developing future treatment regimens for the treatment of addiction.

## Future considerations for how interactions between the D1-like dopamine receptor system and the dynorphin/KOR system mediate pair bond maintenance

Following the establishment of a pair bond, male prairie voles show an enhancement in KOR-agonist induced decreases in stimulated DA within the NAc shell, despite having an overall reduction in KOR binding density within this region. Given the presumption that a reduction in receptor number would also reduce the efficacy of an agonist to produce the measured physiological response, these findings appear to contradict one another. However, G protein-coupled receptors (GPCRS), including KORs, are dynamic proteins that can adopt multiple conformational states, resulting in variability in ligand binding affinity as well as efficacy of the receptor to activate distinct downstream signaling cascades (*Bruchas and Chavkin, 2010*). For example, G proteins have been shown to pre-couple with receptors (*Nobles et al., 2005*) and this coupling can lead to conformational changes in the extracellular portion of the receptor that enhance affinity of the ligand for the receptor (*Yan et al., 2008*). Moreover, the percent of G protein receptor coupling can vary as a function of receptor density (*Yan et al., 2008*). Thus, under certain physiological conditions, it is possible for receptor coupling efficiencies to be enhanced, resulting in a fewer number of receptors to be required for an agonist to produce the maximal biological response (*Kenakin, 2002*).

With binding of the agonist, changes in conformational state of the receptor can also be induced, to increase affinity of the G protein to the receptor. In addition, GPCRS can engage a diverse array of signaling pathways in a manner that is dependent on the ligand (*White et al., 2014*), cellular milieu (*Yan et al., 2008*), as well as lipid membrane properties (*Nygaard et al., 2013*). Thus, there are wide variety of dynamic receptor states that can influence functional interactions between the ligand and the receptor as well as the receptor and its G protein. Future research is therefore necessary to determine if, in addition to pair bond induced changes in striatal KOR expression patterns, if KOR mediated signaling properties also vary as a function of social state as well as the potential for such alterations to influence DA transmission dynamics. Finally, given that KORs are found on the terminal regions of multiple inputs to the NAc (glutamatergic, GABAergic, and dopaminergic) (*Meshul and McGinty, 2000*; *Svingos et al., 2001*; *Hjelmstad and Fields, 2003*) more work will also be necessary to determine if KOR binding is globally decreased within the NAc of pair bonded males or if this decrease is restricted to a specific sub-population of inputs.

## Conclusion

In the present study, we provide extensive detail of the neural mechanism involved in the maintenance of enduring social bonds. Specifically, we show that neural systems involved in aversive valence processing, such as the dynoprhin/KOR system (*Bals-Kubik et al., 1989*; *Land et al., 2008*; *Chartoff et al., 2009*; *Koob and Volkow, 2010*; *Schindler et al., 2012*; *Al-Hasani et al., 2015*), as well as those involved in the orchestration of socially motivated behavioral states, such as the D1-like receptor system, (*Balfour et al., 2004*; *Champagne et al., 2004*; *Aragona et al., 2009*; *Hull, 2011*; *Chevallier et al., 2012*; *Gunaydin et al., 2014*) interact within the NAc shell to mediate selective aggression and the maintenance of monogamous bonds. Importantly, understanding the neurobiology of social bonding has important translational implications for psychiatric disorders of a social nature as well as motivational/affective disorders (*Feldman, 2015*). Thus, further investigation of social reward processing in the prairie vole model has the potential to reveal how social bonding alters motivational circuitry in a manner that buffers against psychopathology.

# Materials and methods

### Subjects

Subjects were adult prairie voles bred in a laboratory colony at the University of Michigan. Subjects were weaned at 21 days of age and initially housed in same-sex sibling pairs. Animals were housed in a 14 hr light/10 hr dark cycle (lights on at 6 AM and off at 8 PM) and all experiments occurred during the light phase of the animals cycle. Food and water was available *ad libitum* (*Resendez et al., 2012*; *Resendez and Aragona, 2013*).

### Housing conditions

For experiments that required pair bonded prairie voles, adult subjects were paired with an opposite sex partner for 14 days in a large cage that subsequently became the pair's 'home cage' cage. This cohabitation time allows for nest sharing, mating, and impregnation (*Aragona et al., 2006*). Pregnancy was confirmed by extracting embryos from pregnant females and subsequently categorizing pregnancy status by average neonatal weight of the embryos (*Curtis, 2010*; *Resendez et al., 2012*)

### Stereotactic surgery

Subjects were anesthetized with a mixture of ketamine (90 mg/kg) and xylazine (10 mg/kg) administered at 0.1% of total body weight. Stereotactic surgery was subsequently performed to implant a 26-gauge bilateral guide cannula (Plastics One, Roanoke, VA) into the NAc shell (+1.7 mm A/P; ± 1 mm M/L; -4.5 mm D/V) (*Resendez et al., 2012*). Cannulas were secured to the skull with stainless steel screws and dental cement. Following surgery, males were given 0.1 mL ketoprofen and returned to their home cage to recover with either their cage mate or mating partner 3 days prior topartner preference, selective aggression, or conditioned place preference testing.

### Partner preference

Three days prior to behavioral testing, a guide cannula was implanted above the NAc shell (*Resendez and Aragona, 2013*). Immediately prior to pairing with an opposite-sex conspecific, male subjects received site-specific injections of either aCSF or 1 µg U50,488 (KOR agonist) (*Muschamp et al., 2011*). Following injections, subjects cohabitated with a female partner for 1 hr and were next placed in a 3-chambered partner preference apparatus with their partner restricted to one chamber and a novel opposite-sex individual (stranger) restricted to the opposite chamber. Test subjects were free to move throughout the apparatus. The 3 hr test was recorded and later scored by an experimentally blind observer for the duration of time spent in side-by-side contact with either the partner or stranger.

### Resident-intruder tests

1 hr prior to resident-intruder testing, subjects received site-specific infusions of one of the following treatment groups: aCSF, 10 ng SCH 23,390 (D1 receptor antagonist), 10 ng SCH 23,390 and 1 µg U50,488, or 0.4 ng SKF 38,393 (D1 receptor agonist) and 500 µg norBNI (KOR antagonist)

(*Aragona et al., 2006*; *Resendez et al., 2012*). 1 hr after drug infusion, the subject was placed in its home cage (in isolation) and its behavior was recorded for 10 min, allowing time for acclimation to the testing environment and the assessment of locomotor activity. Locomotor activity was assessed by counting the number of cage crosses made during the 10-min habituation period. Next, a same-sex stimulus animal was introduced to the subject's home cage and behavioral interactions were recorded for 10 min. Resident-intruder tests were scored for the frequency of aggressive behaviors (offensive rears, lunges, bites, and chase frequency). The latency to engage in agonistic behavior was determined by the first time point in which an aggressive encounter occurred. If an aggressive encounter was not observed during the testing period, an attack latency of 10 min was applied. Affiliative behavior was assessed by quantifying the sum of the duration of time that the test subject spent investigating and in side-by-side contact with the intruder.

## Conditioned place preference

A non-biased conditioned place preference assay was used to assess the rewarding properties of 1 mg/kg of amphetamine, a dose that has been shown to reliably elicit a preference for the drug paired compartment in this species (*Aragona et al., 2007*; *Liu et al., 2010*). A pre-test was conducted on day 1 of conditioning to determine if the test subject had a bias for either side of the chamber. Test subjects that showed a robust preference for either chamber (greater than 67% of the pre-test) were excluded from the study. On the following 3 days, subjects were administered saline in the preferred chamber and 1 mg/kg AMPH the non-preferred chamber and placed in the chamber for 40 min. The order of injections was counterbalanced between subjects and treatments were administered 6 hr apart. On day five of testing, a preference for the amphetamine paired chamber was determined by placing the subject in the apparatus and allowing it to freely roam either compartment for 30 min.

To determine how social conditioning impacts AMPH reward, male subjects were placed in one of several housing conditions: sibling housed, housed with a novel male, house with an OVX female, or housed with an intact female. To determine if alterations in KOR binding contribute to the protective effects of pair bonding against drug reward, non-paired or paired subjects received either peripheral administration of saline or 50 mg/kg nor-BNI dissolved in sterile saline on day 1 of conditioning. To determine if the impact of KORs on drug reward processing was specific to the NAc shell, test subjects received site-specific microinfusions of either aCSF or 500 μg nor-BNI into the NAc shell on day 1 of conditioning. These doses were chosen because they have previously been shown to reduce selective aggression in male prairie voles without altering locomotor activity (*Resendez et al., 2012*). For males paired with intact females, female partners were checked for successful pregnancy after male subjects completed the post-test on day 5 of testing (*Curtis, 2010*; *Resendez et al., 2012*).

## Measuring mRNA by reverse transcriptase PCR

Tissue punches from the dorsal and ventral striatum were processed for mRNA quantification as previously described (*Day et al., 2013*). Total RNA was extracted using the RNeasy Mini kit (Qiagen) following the manufacturer's instructions. mRNA was reverse transcribed using the iScript RT-PCR kit (Bio-Rad). Specific intron-spanning primers were used to amplify cDNA regions for transcripts of interest (*Drd1, Drd2*, Drd3, *Pdyn, Penk, Oprk1, Oprm1, Oxtr, and Avpr1a). Nadh* was used as an internal control for normalization using the ΔΔCt method within the VS. *Oprm1* was used as the control within the DS because mRNA for this gene did not differ between groups and *Nadh* mRNA levels were found to be affected by pregnancy in females.

## Receptor autoradiography

KOR autoradiography was used to quantify changes in receptor density across the striatum as a function of pair bonding and AMPH administration. To examine the effect of pair bonding on KOR binding density, subjects were either paired with a novel female for two weeks or remained housed with a same-sex sibling. To determine the effects of AMPH exposure on KOR binding density, striatal slices from male prairie voles receiving either control injections of saline or injections of 1 mg/kg AMPH for three days were also processed for KOR autoradiography. Follow the completion of either social or drug exposure, subjects were sacrificed via rapid decapitation 24 hrs following the last day

of pairing (for the pair bonding manipulation) or the final injection (for the AMPH manipulation). Brains were removed, immediately frozen on powdered dry ice, and stored at −80°C until sectioning. A cryostat was used to section the brain into a 1:4 series at 20 µM. Sections were directly mounted onto glass slides and stored at −80°C until time of assay.

Autoradiography for KORs was conducted with $^{3}$H U-69,593 radioligands (Perkin Elmer, USA) (*Resendez et al., 2012*). On the day of the assay, slides were thawed at room temperature (RT) until dry and fixed in 0.1% paraformaldehyde (pH 7.4; 2 min). Slides were then washed in 50 mM Tris buffer (pH 7.4; 10 min; 2 washes) and incubated for 1 hr in tracer buffer containing 50 mM Tris buffer, 10 mM $MgCl_2$, 0.1% bovine serum albumin, and 1 nM $^{3}$H U-69,593 for visualization of KOR binding. Next, slides were washed in 50 mM Tris containing 0.2% $MgCl_2$ for 5 min at 4°C (4 washes) then for 30 min at RT (1 wash). Slides were briefly dipped in nanopure $H_2O$, allowed to dry at RT and then exposed to BAS Imaging Plates (GE Life Sciences, Piscataway, NJ) for 2 weeks. All plates were scanned on the BAS-5000 plate reader using BAS-5000 Image Reader software (Version 1.8). Binding densities were determined with region of interest analysis using ImageJ software.

## FSCV

Following rapid, live decapitation, brains were quickly extracted, immediately submerged in cold, pre-oxygenated high sucrose aCSF consisting of 180 mM sucrose, 30 mM NaCl, 4.5 mM KCl, 1 mM $MgCl_2$, 26 mM $NaHCO_3$, 1.2 $NaH_2PO_4$, and 10 mM D-Glucose in deionized $H_2O$ (pH 7.4), and sectioned into coronal slices (400 µm) containing the DS, the NAc core, and the NAc shell. Following sectioning, slices were transferred to RT aCSF buffer solution consisting of 176.13 mM ascorbate, 180.16 mM glucose, 84.01 mM sodium bicarbonate, 58.44 mM NaCl, 156 mM $NaH_2PO_4$, 74.56 mM KCl, 147.01 mM $CaCl_2$, and 203.30 mM $MgCl_2$ in deionized $H_2O$ (pH 7.4) and incubated for 1 hr. A buffer solution of this same composition (minus ascorbate) was used to perfuse the slices during recordings (1 ml/min). Both buffer solutions were continuously bubbled with 5% $CO_2$ and 95% (*Maina and Mathews, 2010*).

FSCV was conducted with recording electrodes fabricated from 1.2 mm pulled glass capillary tubes, with the carbon fiber cut to approximately 150 µm from the capillary glass seal. Using Tarheel CV (University of North Carolina, Chapel Hill) software written in LABVIEW (National Instruments, Austin, TX), a triangular ramp sweeping from −0.4 V to +1.2 V versus a Ag/AgCl reference was applied to the carbon-fiber electrode at a rate of 10 Hz (*Robinson et al., 2002*). The characteristic oxidation current, seen at +0.6 V during the upward ramp, and reduction current, at -0.2 V during the downward ramp, of DA was identified using a background-subtracted cyclic voltammogram (*Aragona et al., 2009*). The peak currents for DA were converted to concentration by calibrating each electrode to a known concentration of DA (3 µM) (*Sinkala et al., 2012*; *Vander Weele et al., 2014*).

To compare differences in striatal DA release properties between non-paired and pair bonded voles, FSCV was conducted in striatal slice preparations (*Singer et al., 2016*). DA release was evoked by a 1 or 20-pulse stimulation (350 µA) delivered in 5-min increments at 20 Hz with a bipolar stimulating electrode placed on the surface of the striatal slice approximately 150 µm from the recording electrode (*Zhang et al., 2009*). Each recording was 15 s in duration and DA release was evoked at 5 s. A total of 3 recordings at each pulse were made within each region and peak DA release was averaged for each subject. Slice stimulations occurred at regular 5-min intervals and readings were only recorded for experimental purposes once DA release was stabilized (*Calipari et al., 2012*).

## KOR dose response in the NAc shell

FSCV was used to assess changes in DA/KOR interactions following the establishment of a pair bond. The KOR agonist BRL 5237 hydrochloride was bath applied to striatal slice preparations and DA release was measured (*Britt and McGehee, 2008*). Increasing doses (0.001, 0.01, 0.03, 0.1, 0.3, 1, 3, 10, 20 30 µM) of BRL 52,537 hydrochloride were added every 30-min to the slice's aCSF reservoir, perfused at 1 mL/min. Dose response curves were generated using non-linear regression with the bottom set equal to 0 (*Maina and Mathews, 2010*).

## Statistics

Based upon our previous behavioral pharmacology experiments in voles, 5–8 subjects are needed per group to achieve a p-value of <0.05 with 80% power. Therefore, each group contains at least 6–8 subjects. Consistent with established standards in the literature, we used at least 6 subjects for mRNA and receptor autoradiography experiments and at least 3 subjects for slice FSCV experiments, with multiple samples taken per slice. For most experiments, comparisons were made between biological replicates, i.e., comparisons between treatment groups receiving a pharmacological manipulation or between different social conditions. For FSCV experiments, both biological and technical replicates, i.e., repeated measurements from the same coronal slice under identical preparations were also made. To determine whether the data were normally distributed and equivalent in variance, we examined boxplots for each group. In cases where boxplots revealed that the data were not normally distributed or there was a lack of equal variance among groups, nonparametric tests were used. Statistical significance was assessed with either a t-test, one-way ANOVA, or two-way ANOVA. An alpha level was set at $p \leq 0.05$ for all statistical analysis. All analysis were performed in SPSS version 21 for Windows.

## Acknowledgements

This work was supported by NSF Grant 0953106 and Office of Vice President Research Grant (University of Michigan, Ann Arbor) to BJA SLR is supported by the North Carolina Institute for Developmental Disabilities (T32 HD040127). JJD is supported by the National Institute on Drug Abuse (DA034681). We acknowledge Nina Dutta and Julia Tsinberg for assistance with video scoring.

---

## Additional information

### Funding

| Funder | Grant reference number | Author |
|---|---|---|
| National Science Foundation | 0953106 | Brandon J Aragona |
| National Institute of Child Health and Human Development | T32HD040127 | Shanna L Resendez |
| National Institute on Drug Abuse | DA034681 | Jeremy J Day |
| University of Michigan | Office of Vice President Research Grant | Brandon J Aragona |
| National Institute of Mental Health | 5R21MH096216-02 | Brandon J Aragona |

The funders had no role in study design, data collection and interpretation, or the decision to submit the work for publication.

---

### Author contributions

SLR, Conception and design, Acquisition of data, Analysis and interpretation of data, Drafting or revising the article; PCK, Conception and design, Acquisition of data, Drafting or revising the article; JJD, FKM, LNE, AZM, TAM, Conception and design, Acquisition of data, Analysis and interpretation of data; CH, CJA, JWM, MAK, Acquisition of data, Analysis and interpretation of data; KAP-S, BJA, Conception and design, Analysis and interpretation of data, Drafting or revising the article; NN, Acquisition of data, Analysis and interpretation of data, Drafting or revising the article

### Author ORCIDs

Shanna L Resendez, http://orcid.org/0000-0003-3831-5481

### Ethics

Animal experimentation: All experiments in this study were performed in accordance with the recommendations in the Guide for the Care and Use of Laboratory Animals of the National Institutes of Health. All of the animals were handled according to approved institutional animal care and use

committee (IACUC) protocols (#5531) of the University of Michigan. Experiments conducted in this study were approved by the Institutional Biosafety Committee (#1331) at the University of Michigan. All surgery was performed under ketamine and xylazine anesthesia, and every effort was made to minimize suffering.

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
