## [Decision Letter]

Thank you for submitting your article "Motivation and valence processing systems interact to maintain monogamous pair bonds" for consideration by *eLife*. Your article has been reviewed by three peer reviewers, and the evaluation has been overseen by Richard Palmiter as the Reviewing Editor and a Senior Editor. The following individual involved in review of your submission has agreed to reveal their identity: James Burkett (peer reviewer).

The reviewers have discussed the reviews with one another and the Reviewing Editor has drafted this decision to help you prepare a revised submission.

Summary:

This paper describes a series of experiments indicating how interactions between dopamine signaling and kappa-opioid signaling in the nucleus accumbens promotes pair bonding in prairie voles.

Essential revisions:

While there is general enthusiasm for this paper, there are a number of issues raised by the three reviewers that are difficult to condense into a specific set of essential revisions. Most of the comments (see below) reflect specific details of experimental design and/or interpretation rather than a request for more experiments. Thus, the authors should consider all of the points raised and revise their manuscript so that it is more focused and more precise.

Reviewer #1:

This is an interesting and important study. The wide range of approaches used is very impressive: behavioral, mRNA levels, receptor binding autoradiography, gene sequencing, and fast scan cyclic voltammetry are all included. But the study is written in a rambling disconnected style that obscures the points. In particular, too much discussion is incorporated into the Results where speculation is mixed with objective presentation of findings.

The Abstract presents inference and speculation rather than clear statements of findings followed by one line of implication.

The Introduction is too long and needs to more concisely state the question addressed. Similarly, the Results section is written in a distracting conversational style that obscures the point.

Results (second paragraph) "activation of NAc shell KORs" is very vague. Neither the drug nor dose is specified in the paragraph or figure legend. "KOR treated" is sloppy phrasing. The choice of BRL 5237 as the kappa agonist needs to be better justified and citations supporting claims of receptor selectivity need to be provided. Many more studies use the U compounds, thus the choice of BRL makes it more difficult to compare these findings with others. Because the vole kappa receptor is genetically distinct from rat and mouse, the agonist specificity of BRL needs to be verified by antagonist controls, and cannot be assumed. "strangest effects expected to occur in the ventral striatum." is an unfortunate typo.

It is troubling that the significance of the differences shown in Figure 2 is driven by the variance, not the magnitude of the difference. Results shown in Figure 2 seem largely descriptive and no strong conclusions are drawn.

The dose-response curves shown in Figure 7 are not optimally clear. First, it's not clear whether the hydrochloride' salt needs to be specified. Second the concentration units (μM?) are not labeled in either the axis or in the insets (F & G).

The authors state, "The slope of the dose response curve also significantly differed between paired and non-paired males (t test; t(6) = 3.74, p = 0.009, Figure 6)." (actually Figure 7). First, the confidence intervals of the EC50s should be presented. Second, the interpretation of the change in slope is not explained. Typically, DR slopes more shallow than predicted by mass-action kinetics result from drug interaction with a high and low affinity site. The increase in slope may have been caused by the inactivation of a high affinity site. How do the authors interpret this change?

The experiments seem to have loose and superficial connections. The conclusions are plausible, but alternative interpretations are not excluded.

Reviewer #2:

1) The title strongly emphasizes motivation and valence processing systems, as opposed to the dopamine and k-opioid systems. I don't think it's justified to put these psychological terms in the title when they are not the direct focus of the paper. For instance, dopamine plays a role in a great deal more than motivation, and the manuscript is not particularly focused on differentiating motivation from learning or reward.

2) The experiment "Social reward impairs AMPH-induced place conditioning" (Figure 11) divides subjects into a number of groups based on social stimulus. However, there is a major problem with these groups. Five of the groups are experimenter-controlled (Sibling saline, Sibling AMPH, novel male AMPH, OVX female AMPH, intact female AMPH), but the last of these (intact female AMPH) was sub-divided into three self-selected groups (not pregnant, suboptimal, optimal). Mixing experimenter-controlled groups with self-selected groups turns the entire experiment into a quasiexperiment, with additional caveats due to lack of experimental control. The authors are strongly advised to subdivide the intact female AMPH group only for post-hoc analysis purposes. The same results and conclusions are likely to be justified.

3) Throughout the manuscript, the authors draw conclusions about sex-specific effects without having included males and females in the same statistical analysis. Examples: Table 1; Figure 3; the sex difference between bonding-induced elevations in DA release; etc. In most cases, males and females should be combined into a single analysis (mostly by adding a factor to the ANOVA, or using ANOVA instead of t-test) to show sex effects (both main effects and interactions), which should also strengthen the ability to see sex-independent effects.

3) In several places, the authors relied on t-tests to describe their data when ANOVAs would have been more appropriate. Examples: Figure 1 calls for a two-way ANOVA, and the main effect of drug from that ANOVA would provide the same information as Figure 1; the genetic analysis from Table 1 should have been done with a multivariate ANOVA with post-hoc tests; Table 3, same; Figure 6 could be replaced by an ANOVA comparing the two curves.

Reviewer #3:

Prairie voles pair bond with their mating partners and are an important model for studying the neurobiology of social attachment behavior. Resendez et al. have performed a series of studies to understand how signaling through D1-like dopamine receptor (*Drd1*-like) and kappa opioid receptor (KOR) influence pair bonds under a variety of behavioral paradigms, including social familiarity preference, mate guarding, and buffering of preference for amphetamine. These behavioral and pharmacological studies are tied in with analysis of dopamine release in acute slices using fast scan cyclic voltammetry. Despite these impressive wide-ranging studies, no clear picture emerges as to how and where these two signaling systems interact to control pair bonding. This lack of mechanistic insight diminishes my enthusiasm.

I am also puzzled by some of the results. For example, KOR activation leads to preference for an unfamiliar vole (Figure 1) as well as mate guarding or attacking a non-partner, unfamiliar conspecific (Figure 9). How is this to be reconciled? Similarly, KOR binding is reduced in pair bonded voles but activation of KOR reduces dopamine release in these voles. The most parsimonious explanation is that these signaling systems are regulating pair bonding at multiple points in the circuit and also regulating other behaviors. Until these are resolved I don't see how firm conclusions can be drawn.

---

## [Author Response]

Essential revisions:

While there is general enthusiasm for this paper, there are a number of issues raised by the three reviewers that are difficult to condense into a specific set of essential revisions. Most of the comments (see below) reflect specific details of experimental design and/or interpretation rather than a request for more experiments. Thus, the authors should consider all of the points raised and revise their manuscript so that it is more focused and more precise.

Reviewer #1:

This is an interesting and important study. The wide range of approaches used is very impressive: behavioral, mRNA levels, receptor binding autoradiography, gene sequencing, and fast scan cyclic voltammetry are all included. But the study is written in a rambling disconnected style that obscures the points. In particular, too much discussion is incorporated into the Results where speculation is mixed with objective presentation of findings.

The Abstract presents inference and speculation rather than clear statements of findings followed by one line of implication.

We agree with the reviewer and have rewritten the Abstract to highlight the main findings presented in the paper.

The Introduction is too long and needs to more concisely state the question addressed. Similarly, the Results section is written in a distracting conversational style that obscures the point.

The reviewer is correct that the length of the Introduction can be more concisely written and we have therefore cut the length of the Introduction by 1/3. In addition, efforts have been made to reduce the conversational tone of the Results section.

Results (second paragraph) "activation of NAc shell KORs" is very vague. Neither the drug nor dose is specified in the paragraph or figure legend. "KOR treated" is sloppy phrasing.

The specific KOR agonist used and dose of the drug has been added to the Results section (subsection “KORs within the NAc shell encode social aversion”, second paragraph). The language referring to “KOR treated” males has been rephrased throughout the manuscript.

The choice of BRL 5237 as the kappa agonist needs to be better justified and citations supporting claims of receptor selectivity need to be provided. Many more studies use the U compounds, thus the choice of BRL makes it more difficult to compare these findings with others. Because the vole kappa receptor is genetically distinct from rat and mouse, the agonist specificity of BRL needs to be verified by antagonist controls, and cannot be assumed.

While we agree with the reviewer that more work is necessary to understand the differences in KOR pharmacology between prairie voles and other commonly studied species, this specific area of research was not the primary focus of the present study and would require extensive follow up studies that are beyond the scope of the submitted manuscript. The KOR agonist, BRL 5237, was chosen based on a previous study that utilized FSCV in striatal slice preparations collected from rats to examine KOR modulation of DA release within the NAc shell. This study identified a significant reduction in stimulated dopamine release following a 1μM bath application of BRL 5237 (Britt & McGehee, 2008). We included this dose, in addition to others, in our studies to show that a significant reduction in stimulated dopamine release within the nucleus shell of prairie voles required a much higher dose of this agonist. This study also utilized a KOR antagonist to show that the drug is specific to KOR receptors under these conditions. Although other studies since this publication have utilized the U50, 488 KOR agonist in combination with FSCV in striatal slice preparations, when we initially began our studies, many of these papers had not yet been published and we therefore chose the BRL 5237 KOR agonist to allow us to compare our results to other previously published findings. Additionally, in a previous publication, we conducted extensive behavioral pharmacological comparisons utilizing the U50,488 agonist to show that a much higher dose of the KOR antagonist, nor-BNI, was required to block the locomotor and analgesic effects produced by U50,488 in both male and female prairie voles. Thus, the necessity for higher doses of drugs that act at the KOR in this species is consistent across multiple pharmacological compounds as well as both in vivo and in vitro measures. Finally, although our genetic analysis shows that the prairie vole KOR differs from that of rats, mice, humans, as well as guinea pigs, it also shows that the prairie vole KOR is more similar to that of guinea pigs and humans than that of rats and mice. Importantly, the pharmacological compounds utilized in the present study have been shown to be specific in guinea pigs and humans despite having some differences in pharmacological properties of the drugs when compared to rats and mice. See the second paragraph of the subsection “Pair bonding alters KOR regulation of DA transmission in a sex-specific manner” for the amino acids in which the prairie vole KOR differs from other species.

"strangest effects expected to occur in the ventral striatum." is an unfortunate typo.

This typo has been corrected.

It is troubling that the significance of the differences shown in Figure 2 is driven by the variance, not the magnitude of the difference. Results shown in Figure 2 seem largely descriptive and no strong conclusions are drawn.

We appreciate this comment. The RT-qPCR data in Figure 2 shows that in both males and females, pair housing increased the expression of *Pdyn* as well as *Drd1* mRNA within the ventral, but not dorsal, striatum. These significant increases are largely due to the size of the effect, since the variance in expression of these genes is similar between groups. However, as the reviewer points out, there are other transcripts (*Penk* and *Oprk1*) that exhibited a similar magnitude of difference between groups. Yet differences between these groups did not meet the criteria for statistical significance (p < 0.05) due to a larger amount of variability. The specific p values for all non-significant comparisons can be found within Table 1 of the manuscript.

It is important to note that for each sample, RT-qPCR reactions were completed in triplicate and normalized to housekeeping genes as a within-sample control. Additionally, we only observed this degree of variability at these genes, and only in the ventral striatum. Together, this suggests that the variability in measured expression of these genes is biological rather than technical in nature. We interpret these results to indicate that any potential changes in the expression of these genes following pair bonding were not consistent across animals within a group. Finally, while larger sample sizes may be sufficient to detect these more variable effects, we note that the samples sizes used in the present study are more than adequate for this type of experiment, and also that the lack of change in these genes is in no way an essential component of this manuscript.

The dose-response curves shown in Figure 7 are not optimally clear. First, it's not clear whether the hydrochloride' salt needs to be specified. Second the concentration units (uM?) are not labeled in either the axis or in the insets (F & G).

We have removed hydrochloride from the label. The concentration units (μM) have been added to the figure.

The authors state, "The slope of the dose response curve also significantly differed between paired and non-paired males (t test; t(6) = 3.74, p = 0.009, Figure 6)." (actually Figure 7). First, the confidence intervals of the EC50s should be presented. Second, the interpretation of the change in slope is not explained. Typically, DR slopes more shallow than predicted by mass-action kinetics result from drug interaction with a high and low affinity site. The increase in slope may have been caused by the inactivation of a high affinity site. How do the authors interpret this change?

A table including the 95% confidence interval of the IC50s has been included (Table 4). Given that a lower dose of the KOR agonist was needed to produce a 50% decrease in stimulated DA release in pair bonded prairie voles and that this group also had a reduction in KOR binding, we think it is unlikely that changes in slope are related to inactivation of a high affinity binding site because this would result in a reduction in agonist potency. Differences in the shape of the concentration response curve may also reflect mechanistic differences in the binding of the drug and/or in activation of downstream signaling targets. Given that the goal of the experiment was to examine differences in interactions between the DA/and KOR system between pair bonded and non-pair bonded prairie voles, we were not able to determine possible mechanistic differences in KOR function between these two groups in the present study. For this reason, we do not think it is appropriate to discuss possible mechanistic differences in the Results section, but have included such information in the Discussion section “Future considerations for how interactions between the D1-like dopamine receptor system and the dynorphin/KOR system mediate pair bond maintenance”).

The experiments seem to have loose and superficial connections. The conclusions are plausible, but alternative interpretations are not excluded.

We have added transitional sentences between experiments to clarify their connections.

Reviewer #2:

1) The title strongly emphasizes motivation and valence processing systems, as opposed to the dopamine and k-opioid systems. I don't think it's justified to put these psychological terms in the title when they are not the direct focus of the paper. For instance, dopamine plays a role in a great deal more than motivation, and the manuscript is not particularly focused on differentiating motivation from learning or reward.

We agree with the reviewer and have therefore rewritten the title as, “Dopamine and opioid systems interact within the nucleus accumbens to maintain monogamous pair bonds**”**

2) The experiment "Social reward impairs AMPH-induced place conditioning" (Figure 11) divides subjects into a number of groups based on social stimulus. However, there is a major problem with these groups. Five of the groups are experimenter-controlled (Sibling saline, Sibling AMPH, novel male AMPH, OVX female AMPH, intact female AMPH), but the last of these (intact female AMPH) was sub-divided into three self-selected groups (not pregnant, suboptimal, optimal). Mixing experimenter-controlled groups with self-selected groups turns the entire experiment into a quasiexperiment, with additional caveats due to lack of experimental control. The authors are strongly advised to subdivide the intact female AMPH group only for post-hoc analysis purposes. The same results and conclusions are likely to be justified.

We agree with the reviewer and have recombined the data for all pair housed subjects into one group (regardless of pregnancy status of the pair) to be analyzed in a one-way ANOVA. Indeed, when data for paired subjects were combined into a single group (pair housed), the overall group data exhibited protective effects against the rewarding properties of amphetamine. Data were subsequently separated for further comparison to show that within the pair housed group, only males whose females became pregnant during the two-week pairing period exhibited protective effects against the rewarding properties of amphetamine (subsection “D1-like and KORs interact to mediate selective aggression”, last paragraph).

3) Throughout the manuscript, the authors draw conclusions about sex-specific effects without having included males and females in the same statistical analysis. Examples: Table 1; Figure 3; the sex difference between bonding-induced elevations in DA release; etc. In most cases, males and females should be combined into a single analysis (mostly by adding a factor to the ANOVA, or using ANOVA instead of t-test) to show sex effects (both main effects and interactions), which should also strengthen the ability to see sex-independent effects.

To provide direct comparisons between the sexes, additional analyses were conducted on the data when possible. Specifically, Figure 3—figure supplement 1 shows analysis for sex differences in KOR binding density (subsection “Sex specific alterations in KOR binding”, last paragraph) Figure 5—figure supplement 1 shows analysis for sex differences in pair bond induced alterations in dopamine transmission (subsection “Pair bond induced enhancement of DA release”, first paragraph), Figure 6—figure supplement 1 shows sex differences in the influence of fecundity on selective aggression (subsection “Pair fecundity influences DA transmission dynamics in a sexually dimorphic manner”, second paragraph), Figure 7—figure supplement 2 supplement shows sex differences in pair bond induced alterations in KOR modulation of dopamine transmission (subsection “Pair bonding alters KOR regulation of DA transmission in a sex-specific manner”, fourth paragraph), and Figure 10—figure supplement 1 supplement shows sex differences in amphetamine induced alterations in KOR binding density (subsection “Amphetamine-induced neuroplasticity mimics that of pair bonding”, last paragraph).

Unfortunately, we were not able to conduct such analysis on the mRNA data because all data were normalized to a control gene that was processed in parallel with samples from paired and non-paired subjects of each sex to account for small differences in RNA that go into the reaction as well as the efficiency of the cDNA synthesis. Due to the large number of samples that were used for analysis in this study, data from male and female subjects had to be processed separately at different time points and it would therefore not be ideal to combine these data sets for analysis.

3) In several places, the authors relied on t-tests to describe their data when ANOVAs would have been more appropriate. Examples: Figure 1 calls for a two-way ANOVA, and the main effect of drug from that ANOVA would provide the same information as Figure 1; the genetic analysis from Table 1 should have been done with a multivariate ANOVA with post-hoc tests; Table 3 and Table 4, same; Figure 6 could be replaced by an ANOVA comparing the two curves.

The analysis for the data shown in Figure 1 has been changed to a two-way ANOVA (subsection “KORs within the NAc shell encode social aversion”, second paragraph). However, Figure 1 and Figure 1 represent different behavioral measures. Figure 1 shows the duration of time spent in each compartment of the test chamber, while Figure 1 shows the total duration on time spent in direct side-by-side contact with both social stimuli. Moreover, it is possible for the test subject to be in one of the social stimulus chambers, but not in direct contact with the stimulus.

The analysis for the data shown in Figure 3 (subsection “Sex specific alterations in KOR binding”), Figure 10 (subsection “Amphetamine-induced neuroplasticity mimics that of pair bonding”, last paragraph), Table 1, Table 2, Table 3, and Table 5 (previously Table 4) has been changed to a two-way ANOVA. A repeated measures two-way ANOVA was used to analyze the dose response curve data shown in Figure 7 (subsection “Pair bonding alters KOR regulation of DA transmission in a sex-specific manner”) as well as Figure 7—figure supplement 2 (subsection “Pair bonding alters KOR regulation of DA transmission in a sex-specific manner”, fourth paragraph).

For the genetic data, the data for each gene were compared independently because we did not hypothesize that the expression of one gene would be different or have a meaningful relationship with the expression of another gene.

Reviewer #3:

*Prairie voles pair bond with their mating partners and are an important model for studying the neurobiology of social attachment behavior. Resendez et al. have performed a series of studies to understand how signaling through D1-like dopamine receptor (Drd1-like) and kappa opioid receptor (KOR) influence pair bonds under a variety of behavioral paradigms, including social familiarity preference, mate guarding, and buffering of preference for amphetamine. These behavioral and pharmacological studies are tied in with analysis of dopamine release in acute slices using fast scan cyclic voltammetry. Despite these impressive wide-ranging studies, no clear picture emerges as to how and where these two signaling systems interact to control pair bonding. This lack of mechanistic insight diminishes my enthusiasm.*

I am also puzzled by some of the results. For example, KOR activation leads to preference for an unfamiliar vole (Figure 1) as well as mate guarding or attacking a non-partner, unfamiliar conspecific (Figure 9). How is this to be reconciled?

While we agree with the reviewer that it is not possible to definitively conclude that a preference for the unfamiliar partner was formed versus avoidance of the familiar partner, we think that given the sum of our data, it is more likely that pairing a social stimulus with KOR activation produces avoidance behavior. Moreover, even in the case that a preference was formed for the unfamiliar social stimulus, this preference would still require that the familiar partner that was paired with KOR activation to be avoided. Finally, it is well established that animals will avoid stimuli that have previously been paired with an aversive experience and our study design employed similar conditioning procedures. Specifically, prior to pairing with a female partner, males were administered either aCSF or a KOR agonist into the NAc shell. After the one-hr pairing, males were then given the option to choose to spend time in contact with the familiar female partner or a novel stranger. Males who were administered a KOR agonist immediately prior to pairing avoided the female partner that had been paired with KOR activation suggesting that activation of KOR while in the presence of a social stimulus results in the stimulus to be encoded as aversive and avoided. This finding is similar to previous research showing that pairing an odorant with either stress or KOR activation can result in the avoidance of that odorant (Landet al., 2008).

Similarly, KOR binding is reduced in pair bonded voles but activation of KOR reduces dopamine release in these voles. The most parsimonious explanation is that these signaling systems are regulating pair bonding at multiple points in the circuit and also regulating other behaviors. Until these are resolved I don't see how firm conclusions can be drawn.

This is another important point that has been brought up by the reviewer. Although the data appear to be conflicting, much evidence supports that GPCRs are capable of behaving in a dynamic nature that can vary by ligand binding, receptor density, lipid membrane properties, G protein coupling, as well as the specific downstream signaling cascade that is engaged. While we agree that figuring the mechanism underlying how enhanced reduction of dopamine transmission occurs in pair bonded males despite this group also showing a reduction in KOR binding, this would require extensive amounts of follow up experiments that our beyond the main focus of this paper. Here, our main goal was to determine if interactions between D1-like receptors and KORs mediate the maintenance of monogamous social bonds and future experiments will be required to figure out the mechanism of this interaction. We have highlighted this issue in the Discussion section and added an additional section entitled, “Future considerations for how interactions between the D1-like dopamine receptor system and the dynorphin/KOR system mediate pair bond maintenance” to address this important future area of research.